# EigenVI: score-based variational inference with orthogonal function expansions

**Diana Cai**
Flatiron Institute
dcai@flatironinstitute.org

**Chirag Modi**
Flatiron Institute
cmodi@flatironinstitute.org

**Charles C. Margossian**
Flatiron Institute
cmargossian@flatironinstitute.org

**Robert M. Gower**
Flatiron Institute
rgower@flatironinstitute.org

**David M. Blei**
Columbia University
david.blei@columbia.edu

**Lawrence K. Saul**
Flatiron Institute
lsaul@flatironinstitute.org

## Abstract

We develop EigenVI, an eigenvalue-based approach for black-box variational inference (BBVI). EigenVI constructs its variational approximations from orthogonal function expansions. For distributions over $\mathbb{R}^D$, the lowest order term in these expansions provides a Gaussian variational approximation, while higher-order terms provide a systematic way to model non-Gaussianity. These approximations are flexible enough to model complex distributions (multimodal, asymmetric), but they are simple enough that one can calculate their low-order moments and draw samples from them. EigenVI can also model other types of random variables (e.g., nonnegative, bounded) by constructing variational approximations from different families of orthogonal functions. Within these families, EigenVI computes the variational approximation that best matches the score function of the target distribution by minimizing a stochastic estimate of the Fisher divergence. Notably, this optimization reduces to solving a minimum eigenvalue problem, so that EigenVI effectively sidesteps the iterative gradient-based optimizations that are required for many other BBVI algorithms. (Gradient-based methods can be sensitive to learning rates, termination criteria, and other tunable hyperparameters.) We use EigenVI to approximate a variety of target distributions, including a benchmark suite of Bayesian models from `posteriordb`. On these distributions, we find that EigenVI is more accurate than existing methods for Gaussian BBVI.

## 1 Introduction

Probabilistic modeling is a cornerstone of modern data analysis, uncertainty quantification, and decision making. A key challenge of probabilistic inference is computing a target distribution of interest; for instance, in Bayesian modeling, the goal is to compute a posterior distribution, which is often intractable. Variational inference (VI) [5, 21, 45] is a popular method for scalable probabilistic inference that has worked across a range of applications. The idea behind VI is to approximate the target distribution by the closest member of some tractable family.

One major focus of research is to develop *black-box* algorithms for variational inference [6, 15, 23, 27, 32, 37, 40, 44, 46]. Algorithms for black-box variational inference (BBVI) can be used to

38th Conference on Neural Information Processing Systems (NeurIPS 2024).

approximate any target distribution that is differentiable and computable up to some multiplicative (normalizing) constant; as such, they are extremely flexible. These algorithms have been widely implemented in popular probabilistic programming languages, and they are part of the modern toolbox for practitioners in computational statistics and data analysis [1, 4, 7, 13, 43].

Traditionally, the variational approximations in BBVI are optimized by minimizing the Kullback-Leibler (KL) divergence between the variational family and the target (equivalently, maximizing the ELBO). This strategy is powerful and scalable, but it relies on stochastic gradient descent (SGD), which can be difficult to tune [10, 11, 51]. These difficulties can be acute even for Gaussian variational approximations [27, 40], particularly if these approximations employ full covariance matrices.

More recently, researchers have proposed algorithms for Gaussian BBVI that do not require the use of SGD [6, 37]. Instead of minimizing the KL divergence, these methods aim to match the *scores*, or the gradients of the log densities, between the variational distribution and the target density. These methods exploit the special form of Gaussian distributions to derive closed-form proximal point updates for score-matching. These updates are as inexpensive as SGD, but not as brittle. They show that score-based BBVI can be applied in an elegant way to Gaussian variational families.

In this paper, we show that score-based BBVI also yields simple, closed-form updates for a much broader family of variational approximations. Specifically, we propose a new class of variational families constructed from *orthogonal function expansions* and inspired by solutions to the Schrödinger equation in quantum mechanics. These families are expressive enough to parameterize a wide range of target distributions; at the same time, the distributions in these families are sufficiently tractable that one can calculate low-order moments and draw samples from them. In this paper, we mostly use orthogonal function expansions to construct distributions supported on $\mathbb{R}^D$; in this case, the lowest-order term in the expansion is sufficient to model Gaussian behavior, while higher-order terms account for increasing amounts of non-Gaussianity. More generally, we also show how different basis sets of orthogonal functions can be used to construct variational families over other spaces.

To optimize over a variational family from this class, we minimize an estimate of the Fisher divergence, which measures the scores of the variational distribution against those of the target distribution. We show that this optimization reduces to a minimum eigenvalue problem, thus avoiding the need for gradient-based methods. For this reason, we call our approach *EigenVI*.

We study EigenVI with a variational family constructed from weighted Hermite polynomials. We first demonstrate the expressiveness of this family on a variety of multimodal, asymmetric, and heavy-tailed distributions. We then use EigenVI to approximate a diverse collection of non-Gaussian target distributions from `posteriordb` [35], a benchmark suite of Bayesian hierarchical models. On these problems, EigenVI provides more accurate posterior approximations than leading implementations of Gaussian BBVI based on KL minimization and score-matching.

The organization of this paper is as follows. In Section 2 we introduce the variational families that arise from orthogonal function expansions, and we show how score-matching in these families reduces to an eigenvalue problem. In Section 3 we review the literature related to EigenVI. In Section 4, we evaluate EigenVI on a variety of synthetic and real-data targets. Finally, in Section 5, we discuss limitations and future work.

## 2 Score-based variational inference with orthogonal function expansions

In this section we use orthogonal function expansions to develop new variational families for approximate probabilistic inference. In Section 2.1, we review the basic properties of these expansions. In Section 2.2, we introduce a score-based divergence for VI with these families; notably, for this divergence, the optimization for VI reduces to an eigenvalue problem. Finally in Section 2.3, we consider how to use these variational approximations for unstandardized distributions; in these settings we must carefully manage the trade-off between expressiveness and computational cost.

### 2.1 Orthogonal function expansions

Let $\mathcal{Z} \subseteq \mathbb{R}^D$ denote the support of the target distribution $p$. Suppose that there exists a complete set of orthonormal basis functions $\{\phi_k(z)\}_{k=1}^{\infty}$ on this set. By *complete*, we mean that any sufficiently well-behaved function $f : \mathcal{Z} \to \mathbb{R}$ can be approximated, to arbitrary accuracy, by a particular weighted

Table 1: Examples of orthogonal function expansions in one dimension. The basis functions in the table are not normalized, but they can be rescaled so that their squares integrate to one.

| support | orthogonal family | basis functions $\phi_k(\cdot)$ |
|---|---|---|
| $z \in [-1, 1]$ | Legendre polynomials | $\{1,\ z,\ 3z^2-1,\ 5z^3-3z, \ldots\}$ |
| $z = e^{i\theta} \in S^1$ | Fourier basis | $\{1, \cos\theta, \sin\theta, \cos 2\theta, \sin 2\theta, \ldots\}$ |
| $z \in [0, \infty)$ | weighted Laguerre polynomials | $e^{-\frac{z}{2}}\{1, 1-z, z^2-4z+2, \ldots\}$ |
| $z \in \mathbb{R}$ | weighted Hermite polynomials | $e^{-\frac{z^2}{4}}\{1, z, (z^2-1), (z^3-3z), \ldots\}$ |

sum of these basis functions, and by *orthonormal*, we mean that the basis functions satisfy

$$\int \phi_k(z)\phi_{k'}(z)\,dz = \left\{ \begin{array}{ll} 1 & \text{if } k = k', \\ 0 & \text{otherwise,} \end{array} \right. \tag{1}$$

where the integral is over $\mathcal{Z}$. Define the $K^{\text{th}}$-order variational family $\mathcal{Q}_K$ to be the set containing all distributions of the form

$$q(z) = \left( \sum_{k=1}^{K} \alpha_k \phi_k(z) \right)^2 \quad \text{where} \quad \sum_{k=1}^{K} \alpha_k^2 = 1, \tag{2}$$

and where $\alpha_k \in \mathbb{R}$ for $k = 1, \ldots, K$ are the parameters of the family $\mathcal{Q}_K$. In words, $\mathcal{Q}_K$ contains all distributions that can be obtained by taking weighted sums of the first $K$ basis functions and then *squaring* the result.

Eq. 2 involves a squaring operation, a sum-of-squares constraint, and a weighted sum. The squaring operation ensures that the density functions in $\mathcal{Q}_K$ are nonnegative (i.e., with $q(z) \geq 0$ for all $z \in \mathcal{Z}$), while the sum-of-squares constraint ensures that they are normalized:

$$\int q(z)\,dz = \int \left( \sum_{k=1}^{K} \alpha_k \phi_k(z) \right)^2 dz = \int \sum_{k,k'=1}^{K} \alpha_k \alpha_{k'} \phi_k(z)\phi_{k'}(z)\,dz = \sum_{k=1}^{K} \alpha_k^2 = 1. \tag{3}$$

The weighted sum in Eq. 2 bears a superficial similarity to a mixture model, but note that neither the basis functions $\phi_k(z)$ nor the weights $\alpha_k$ in Eq. 2 are constrained to be nonnegative. Distributions of this form arise naturally in physics from the quantum-mechanical *wave functions* that satisfy Schrödinger's equation [16]. In that setting, though, it is typical to consider complex-valued weights and basis functions, whereas here we only consider real-valued ones.

The simplest examples of orthogonal function expansions arise in one dimension. For example, functions on the interval $[-1, 1]$ can be represented as weighted sums of Legendre polynomials, while functions on the unit circle can be represented by Fourier series of sines and cosines; see Table 1. Distributions on unbounded intervals can also be represented in this way. On the real line, for example, we may consider approximations of the form in Eq. 2 where

$$\phi_{k+1}(z) = \left( \sqrt{2\pi}k! \right)^{-\frac{1}{2}} \left( e^{-\frac{1}{2}z^2} \right)^{\frac{1}{2}} \mathrm{H}_k(z), \tag{4}$$

and $\mathrm{H}_k(z)$ are the *probabilist's* Hermite polynomials given by

$$\mathrm{H}_k(z) = (-1)^k e^{\frac{z^2}{2}} \frac{d^k}{dz^k} \left[ e^{-\frac{z^2}{2}} \right]. \tag{5}$$

Note how the lowest-order basis function $\phi_1(z)$ in this family gives rise (upon squaring) to a Gaussian distribution with zero mean and unit variance.

Figure 1 shows how various multimodal distributions with one-dimensional support can be approximated by computing weighted sums of basis functions and squaring their result. We emphasize that *the more basis functions in the sum, the better the approximation*.

Orthogonal function expansions in one dimension are also important because their Cartesian products can be used to generate orthogonal function expansions in higher dimensions. For example, we can

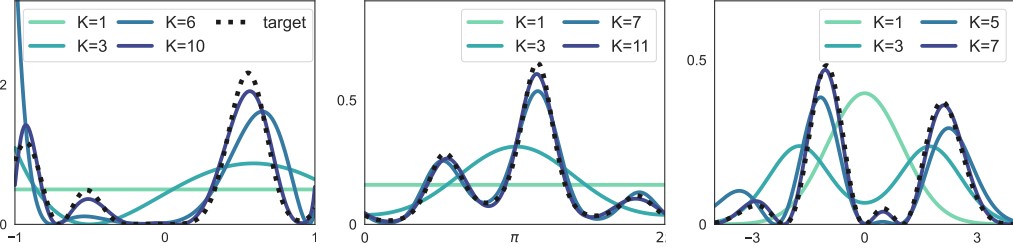

(a) Legendre polynomial expansion     (b) Fourier series expansion     (c) Hermite polynomial expansion

Figure 1: Target probability distributions (black dashed curves) on the interval $[-1, 1]$ (left), the unit circle (middle), and the real line (right), and their approximations by orthogonal function expansions from different families and of different orders; see Eq. 2 and Table 1.

approximate distributions over (say) $\mathbb{R}^3$ by

$$q(z_1, z_2, z_3) = \left( \sum_{i=1}^{K_1} \sum_{j=1}^{K_2} \sum_{k=1}^{K_3} \beta_{ijk} \, \phi_i(z_1) \phi_j(z_2) \phi_k(z_3) \right)^2 \quad \text{where} \quad \sum_{ijk} \beta_{ijk}^2 = 1, \qquad (6)$$

where $\beta_{ijk} \in \mathbb{R}$ now parametrize the family. Note that there are a total $K_1 K_2 K_3$ parameters in the above expansion, so that this method of Cartesian products does not scale well to high dimensions if multiple basis functions are used per dimension. Note that the same strategy can also be used for random variables of mixed type: for example, from Table 1, we can create a variational family of distributions over $\mathbb{R} \times [-1, 1] \times [0, \infty)$ from the Cartesian product of orthogonal function expansions involving Hermite, Legendre, and Laguerre polynomials.

As shown in Figure 1, the approximating distributions from $K^{\text{th}}$-order expansions can model the presence of multiple modes as well as many types of asymmetry, and this expressiveness also extends to higher dimensions. Nevertheless, it remains tractable to sample from these distributions and even to calculate (analytically) their low-order moments, as we show in Appendices A and B.

For concreteness, consider the distribution over $\mathbb{R}^3$ in Eq. 6. The marginal distribution $q(z_1)$ is

$$q(z_1) = \int q(z_1, z_2, z_3) \, dz_2 \, dz_3 = \sum_{ii'} \left[ \sum_{jk} \beta_{ijk} \beta_{i'jk} \right] \phi_i(z_1) \phi_{i'}(z_1), \qquad (7)$$

and from this expression, moments such as $\mathbb{E}[z_1]$ and $\text{Var}[z_1]$ can be calculated by evaluating integrals involving the elementary functions in Table 1. (In practice, these integrals are further simplified by recursion relations that relate basis functions of different orders; we demonstrate how to compute the first two moments for the normalized Hermite family in Eqs. B.19 and B.22.)

To generate samples $\{z^{(t)}\}$, each dimension is sampled as follows: we draw $z_1^{(t)} \sim q(z_1)$ by computing the cumulative distribution function (CDF) of this marginal distribution and then numerically inverting this CDF. Finally, extending these ideas, we can calculate higher-order moments and obtain joint samples via the nested draws

$$z_1^{(t)} \sim q(z_1), \quad z_2^{(t)} \sim q(z_2 \,|\, z_1), \quad z_3^{(t)} \sim q(z_3 \,|\, z_1, z_2). \qquad (8)$$

The overall complexity of these procedures scales no worse than quadratically in the number of basis functions in the expansion. These extensions are discussed further in Appendices A and B.

## 2.2   EigenVI

In variational inference, we posit a parameterized family of approximating distributions and then compute the particular approximation in this family that is closest to a target distribution of interest. Eq. 2 constructs a variational family $\mathcal{Q}_K$ from the orthogonal functions $\{\phi_k(z)\}_{k=1}^K$ whose variational parameters are the weights $\{\alpha_k\}_{k=1}^K$. We now derive *EigenVI*, a method to find $q \in \mathcal{Q}_K$ that is close to the target distribution $p(z)$.

We first define the measure of closeness that we will minimize. EigenVI measures the quality of an approximate density by the *Fisher divergence* [18],

$$\mathscr{D}(q,p) = \int \|\nabla \log q(z) - \nabla \log p(z)\|^2 \, q(z)dz, \tag{9}$$

where $\nabla \log q(z)$ and $\nabla \log p(z)$ are the score functions of the variational approximation and target, respectively. Suppose that $q$ and $p$ have the same support; then the Fisher divergence vanishes if and only if the scores of $q$ and $p$ are everywhere equal.

Though $p$ is, by assumption, intractable to compute, in many applications it is possible to efficiently compute the score $\nabla \log p$ at any point $z \in \mathcal{Z}$. For example, in Bayesian models the score of the target posterior is equal to the gradient of the log joint. This observation is the main motivation for score-based methods in probabilistic modeling [6, 31, 37, 48].

Here we seek the $q \in \mathcal{Q}_K$ that minimizes $\mathscr{D}(q,p)$. But now a challenge arises: it is generally difficult to evaluate the integral for $\mathscr{D}(q,p)$ in Eq. 9, let alone to minimize it as a function of $q$. While it is possible to sample from the distribution $q$, it is not straightforward to simultaneously sample from $q$ and optimize over the variational parameters $\{\alpha_k\}_{k=1}^K$ in terms of which it is defined. Instead, we construct an unbiased estimator of $\mathscr{D}(q,p)$ by importance sampling, which also decouples the sampling distribution from the optimization. Let $\{z^1, z^2, \ldots z^B\}$ denote a batch of $B$ samples drawn from some proposal distribution $\pi$ on $\mathcal{Z}$. From these samples we can form the unbiased estimator

$$\widehat{\mathscr{D}}_\pi(q,p) = \sum_{b=1}^B \frac{q(z^b)}{\pi(z^b)} \left\| \nabla \log q(z^b) - \nabla \log p(z^b) \right\|^2. \tag{10}$$

This estimator should be accurate for appropriately broad proposal distributions and for sufficiently large batch sizes. We can therefore attempt to minimize Eq. 10 in place of Eq. 9.

Now we show that the minimization of Eq. 10 over $q \in \mathcal{Q}_K$ simplifies to a minimum eigenvalue problem for the weights $\{\alpha_k\}_{k=1}^K$. To obtain the eigenvalue problem, we substitute the orthogonal function expansion in Eq. 2 into Eq. 10 for the unbiased estimator of $\mathscr{D}(q,p)$. As an intermediate step, we differentiate Eq. 2 to obtain the scores

$$\nabla \log q(z^b) = \frac{2 \sum_k \alpha_k \nabla \phi_k(z^b)}{\sum_k \alpha_k \phi_k(z^b)}. \tag{11}$$

Further substitution of the scores provides the key result behind our approach: the unbiased estimator in Eq. 10 is a simple quadratic form in the weights $\alpha := [\alpha_1, \ldots, \alpha_K]^\top$ of the orthogonal function expansion,

$$\widehat{\mathscr{D}}_\pi(q,p) = \alpha^\top M \alpha, \tag{12}$$

where the coefficients of the quadratic form are given by

$$M_{jk} = \sum_{b=1}^B \frac{1}{\pi(z^b)} \left[ 2\nabla \phi_j(z^b) - \phi_j(z^b) \nabla \log p(z^b) \right] \cdot \left[ 2\nabla \phi_k(z^b) - \phi_k(z^b) \nabla \log p(z^b) \right]. \tag{13}$$

Note that the elements of the $K \times K$ symmetric matrix $M$ capture all of the dependence on the batch of samples $\{z^b\}_{b=1}^B$, the scores of $p$ and $q$ at these samples, and the choice of the family of orthogonal functions. Next we minimize the quadratic form in Eq. 12 subject to the sum-of-squares constraint $\sum_k \alpha_k^2 = 1$ in Eq. 2. In this way we obtain the eigenvalue problem [8]

$$\min_{q \in \mathcal{Q}_K} \left[ \widehat{\mathscr{D}}_\pi(q,p) \right] = \min_{\|\alpha\|=1} \left[ \alpha^\top M \alpha \right] =: \lambda_{\min}(M), \tag{14}$$

where $\lambda_{\min}(M)$ is the minimal eigenvalue of $M$, and the optimal weights are given (up to an arbitrary sign) by its corresponding eigenvector; see Appendix C for a proof. EigenVI solves Eq. 14.

We note that the eigenvalue problem in EigenVI arises from the curious alignment of three particular choices—namely, (i) the choice of variational family (based on orthogonal function expansions), (ii) the choice of divergence (based on score-matching), and (iii) the choice of estimator for the divergence (based on importance sampling). The simplicity of this eigenvalue problem stands in contrast to the many heuristics of gradient-based optimizations—involving learning rates, terminating criteria, and perhaps other algorithmic hyperparameters—that are typically required for ELBO-based

BBVI [10, 11]. But EigenVI is also not entirely free of heuristics; to compute the estimator in Eq. 10 we must also specify the proposal distribution $\pi$ and the number of samples $B$; see Appendix D for a discussion.

The size of the eigenvalue problem in EigenVI is equal to the number of basis functions $K$ in the orthogonal function expansion of Eq. 2. The eigenvalue problem also generalizes to orthogonal function expansions that are formed from Cartesian products of one-dimensional families, but in this case, if multiple basis functions are used per dimension, then the overall basis size grows exponentially in the dimensionality. Thus, for example, the eigenvalue problem would be of size $K_1 K_2 K_3$ for the approximation in Eq. 6, as can be seen by "flattening" the tensor of weights $\beta$ in Eq. 6 into the vector of weights $\alpha = \mathbf{vec}(\beta)$ in Eq. 2. Finally, we note that EigenVI only needs to compute the minimal eigenvector of $M$ in Eq. 14, and therefore it can benefit from specialized routines that are much less expensive than a full diagonalization.

## 2.3  EigenVI in $\mathbb{R}^D$: the Hermite family and standardization

We now discuss the specific case of EigenVI for $\mathcal{Z} = \mathbb{R}^D$ with the Hermite-based variational family in Eq. 4. For this case, we propose a transformation of the domain that serves to precondition or *standardize* the target distribution before applying EigenVI. While this standardization is not required to use EigenVI, it helps to reduce the number of basis functions needed to approximate the target, leading to a more computationally efficient procedure. It also suggests natural default choices for the proposal distribution $\pi$ in Eq. 10.

Recall that the eigenvalue problem grows linearly in size with the number of basis functions. Before applying EigenVI, our goal is therefore to transform the domain in a way that reduces the number of basis functions needed for a good approximation. To meet this goal for distributions over $\mathbb{R}^D$, we observe that the lowest-order basis function of the Hermite family in Eq. 4 yields (upon squaring) a standard multivariate Gaussian, with zero mean and unit covariance. Intuitively, we might expect the approximation of EigenVI to require fewer basis functions if the statistics of the target distribution nearly match those of this lowest-order basis function. The goal of standardization is to achieve this match, to whatever extent possible, by a suitable transformation of the underlying domain. Having done so, EigenVI in $\mathbb{R}^D$ can then be viewed as a systematic framework to model non-Gaussian effects via a small number of higher-order terms in its orthogonal function expansion.

Concretely, we consider a linear transformation of the domain:

$$\tilde{z} = \Sigma^{-\frac{1}{2}}(z - \mu), \tag{15}$$

where $\mu$ and $\Sigma$ are estimates of the mean and covariance obtained from some other algorithm (e.g., a Laplace approximation, Gaussian variational inference, Monte Carlo, or domain-specific knowledge). We then apply the EigenVI to fit a $K$th-order variational approximation $\tilde{q}(\tilde{z})$ to the target distribution $\tilde{p}(\tilde{z})$ that is induced by this transformation; afterwards, we reverse the change-of-variables to obtain the final approximation to $p(z)$, i.e.,

$$q(z) = \tilde{q}(\tilde{z})|\Sigma|^{-1/2}. \tag{16}$$

Figure 2 shows why it is more difficult to approximate distributions that are badly centered or poorly scaled. The left panel shows the effect of translating a standard Gaussian *away* from the origin and *shrinking* its variance; note how a comparable approximation to the uncentered Gaussian now requires a 16th-order expansion. On the other hand, after standardization, the target can be perfectly fitted by the base distribution in the orthogonal family of reweighted Hermite polynomials. The right panel shows the similar effect of translating the mixture distribution in Figure 1 (right panel); comparing these panels, we see that twice as many basis functions ($K = 14$ versus $K = 7$) are required to provide a comparable fit of the uncentered mixture.

Finally, we note another benefit of standardizing the target before fitting EigenVI; when the target has nearly zero mean and unit covariance, it becomes simpler to identify natural choices for the proposal distribution $\pi$. Intuitively, in this case, we want a proposal distribution that has the same mean but heavier tails than a standard Gaussian. In our experiments, we use two types of centered proposal distributions—uniform and isotropic Gaussian—whose variances are greater than one.

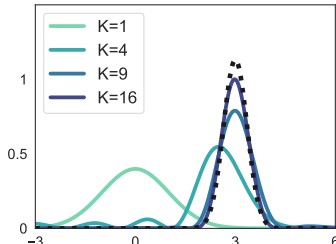 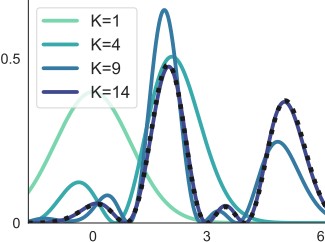

(a) Gaussian target, mean 3 and variance $\frac{1}{8}$    (b) Mixture target (translation of Figure 1c)

Figure 2: Higher-order expansions may be required to approximate target distributions (black) that are not standardized. *Left:* approximation of a non-standardized Gaussian. *Right:* approximation of the mixture distribution in Figure 1 after translating its largest modes away from the origin.

## 3 Related work

Several recent works have considered BBVI methods based on score-matching. These methods take a particularly simple form for Gaussian variational families [6, 37]. The Fisher divergence [18] has been previously studied as a divergence for variational inference [47]. Yu and Zhang [48] propose minimizing a Fisher divergence for semi-implicit (non-Gaussian) variational families; the divergence is minimized using gradient-based optimization. In another line of work, Zhang et al. [50] consider variational families of energy-based models and derive a closed-form solution to minimize the Fisher divergence in this setting.

More generally, there have many studies of VI with non-Gaussian variational families. One common extension is to consider families of mixture models [14, 17, 36]; these are typically optimized via ELBO maximization. BBVI algorithms have also been derived for more expressive variational families of energy-based models [9, 22, 28, 29, 52, 53] and normalizing flows [3, 24, 25, 34, 39, 41]. However the performance of these models, especially the normalizing flows, is often sensitive to the hyperparameters of the flow architecture and optimizer, as well as the parameters of the base distribution [2, 11]. Other aspects of these variational approximations are also less straightforward; for example, one cannot compute their low-order moments, and one cannot easily evaluate or draw samples from the densities of energy-based models.

The variational approximation in EigenVI is based on the idea of squaring a weighted sum of basis functions. Probability distributions of this form arise most famously in quantum mechanics [16]. This idea has also been used to model distributions in machine learning, though not quite in the way proposed here. Novikov et al. [38] propose a tensor train-based model for density estimation, but they do not consider orthogonal basis sets. Similarly, Loconte et al. [33] obtain distributions by squaring a mixture model with negative weights, and they study this model in conjunction with probabilistic circuits. By contrast in this work, we consider this idea in the context of variational inference, and we focus specifically on the use of orthogonal function expansions, which have many simplifying properties; additionally, the specific objective we optimize leads to a minimum eigenvalue problem.

## 4 Experiments

We evaluate EigenVI on 9 synthetic targets and 8 real data targets. In these experiments we use the orthogonal family induced by *normalized Hermite polynomials* (see Table 1), whose lowest-order expansion is Gaussian. Thus, this variational family can model non-Gaussian behavior with the higher-order functions in its basis. We first study 2D synthetic targets and use them to demonstrate the expressiveness of these higher-order expansions. Next, we experiment with target distributions where we systematically vary the tail heaviness and amount of skew. Finally, we apply EigenVI to a set of hierarchical Bayesian models from real-world applications and benchmark its performance against other Gaussian BBVI algorithms.

### 4.1 2D synthetic targets

We first demonstrate how higher-order expansions of the variational family yield more accurate approximations on a range of 2D non-Gaussian target distributions (Figure 3); see Appendix E.2

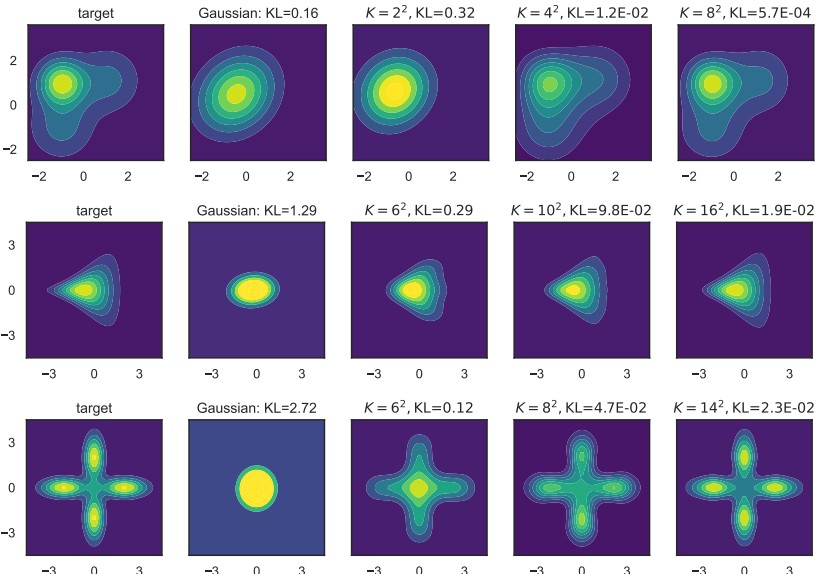

Figure 3: 2D target functions (column 1): a 3-component Gaussian mixture distribution (row 1), a funnel distribution (row 2), and a cross distribution (row 3). We report the $\text{KL}(p; q)$ for the resulting optimal variational distributions obtained using score-based VI with a Gaussian variational family (column 2) and the EigenVI variational family (columns 3–5), where $K = K_1 K_2$.

for details. We report an estimate of $\text{KL}(p; q)$ above each variational approximation. The Gaussian variational approximation is fit using batch and match VI [6], which minimizes a score-based divergence. For EigenVI, the target distributions were not standardized before fitting EigenVI (we compare the costs of the methods in Figure E.1), and the total number of basis functions is $K = K_1 K_2$.

## 4.2 Non-Gaussianity: varying skew and tails in the sinh-arcsinh distribution

We now consider the sinh-arcsinh normal distribution [19, 20], which is induced by transforming a multivariate Gaussian using parameters that control the amount of skew and the weight of the tails. We construct several targets $(D = 2, 5)$ of increasing amounts of non-Gaussianity in the skew or the tails of the distribution, and we refer to these targets as *slight skew and tails*, *more skew and tails*, and *slight skew and heavier tails*; see Appendix E.3 for details. In Figure 4a, we visualize the 2D targets and the EigenVI fits along with their forward KLs. Before applying EigenVI, we standardize the target using a mean and covariance estimated from batch and match VI [6]. In Figure 4b, we measure the EigenVI forward KL under varying numbers of samples $B$ and across increasing numbers of basis functions, given by $K = \prod_{d=1}^{D} K_d$. We also present the forward KL resulting from batch and match VI (BaM) and automatic differentiation VI (ADVI), which both use Gaussian variational families and are run using the same budget in terms of number gradient evaluations. Next we consider similar targets with $D = 5$, which are visualized in in Figure E.2, along with the resulting EigenVI variational approximations. In Figure 4c, we observe greater differences in the number of importance samples needed to lead to good approximations, especially as the number of basis functions increase.

## 4.3 Hierarchical modeling benchmarks from posteriordb

We now evaluate EigenVI on a set of hierarchical Bayesian models [7, 35, 42], which are summarized in Table E.1. The goal is posterior inference: given data observations $x_{1:N}$, the posterior of $z$ is

$$p(z \,|\, x_{1:N}) \propto p(z)p(x_{1:N} \,|\, z) =: \rho(z), \tag{17}$$

where $p(z)$ is the prior and $p(x_{1:N} \,|\, z)$ denotes the likelihood.

We compare EigenVI to 1) automatic differentiation VI (ADVI) [27], which maximizes the ELBO over a full-covariance Gaussian family (ADVI), 2) Gaussian score matching (GSM) [37], a score-based BBVI approach with a full-covariance Gaussian family, and 3) batch and match VI (BaM) [6], which

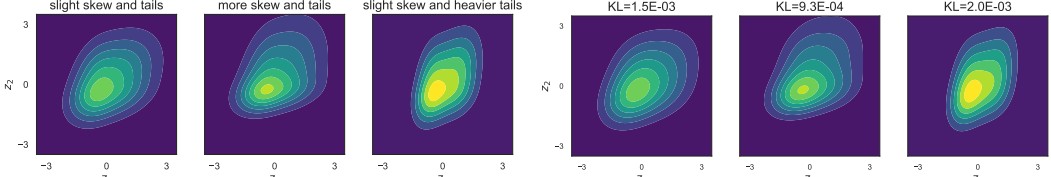

(a) Example 2D targets (left) varying the skew $s$ or tail weight $\tau$ components and their EigenVI fits (right).

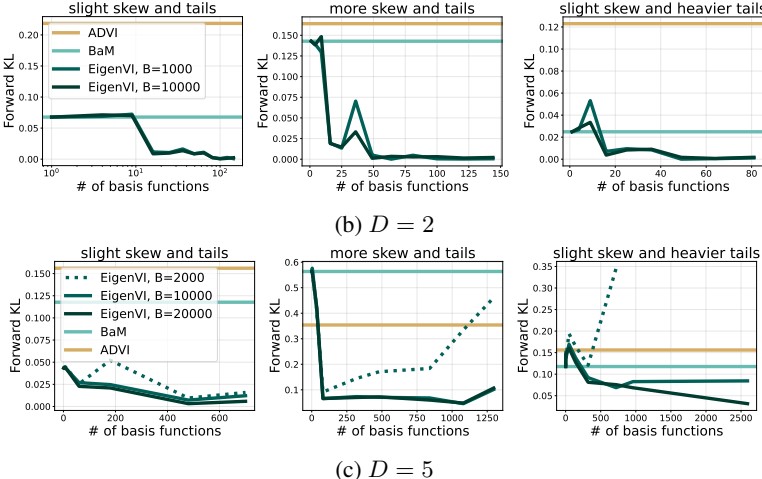

(b) $D = 2$

(c) $D = 5$

Figure 4: Sinh-arcsinh normal distribution synthetic target. Panel (a) shows the three targets we consider in 2D, and their resulting EigenVI fit. Panel (b) shows measures $\mathrm{KL}(p; q)$ for $D = 2$, and panel (c) shows $\mathrm{KL}(p; q)$ for $D = 5$; the $x$-axis shows the number of basis functions, $K = \prod_d K_d$.

minimizes a regularized score-based divergence over a full-covariance Gaussian family. In these examples, we standardize the target using either GSM or BaM before applying EigenVI.

In these models, we do not have access to the target distribution, $p(z \mid x_{1:N})$, only the unnormalized target $\rho$. Thus, we cannot evaluate an estimate of the forward KL. Instead, to evaluate the fidelity of the fitted variational distributions, we compute the empirical Fisher divergence using reference samples from the posterior obtained via Hamiltonian Monte Carlo (HMC):

$$\frac{1}{S} \sum_{s=1}^{S} \|\nabla \log \rho(z^s) - \nabla \log q(z^s)\|^2, \quad z^s \sim p(z \mid x_{1:N}). \tag{18}$$

Note that this measure is not the objective that EigenVI minimizes; it is analogous to the forward KL divergence, as the expectation is taken with respect to $p$. We report the results in Figure 5, computing the Fisher divergence for EigenVI with increasing numbers of basis functions. We typically found that with more basis functions, the scores becomes closer to that of the target.

Finally, we provide a qualitative comparison with real-NVP normalizing flows (NFs) [12], a flexible variational family that is fit by minimizing the reverse KL. We found that after tuning the batch-size and learning rate, NFs generally had a suitable fit. We visualize the posterior marginals for a subset of dimensions from `8schools` in the top three rows, comparing EigenVI, the NF, and BaM. Here, we observe that the Gaussian struggles to fit the tails of this target distribution. On the other hand, EigenVI provides a competitive fit to the normalizing flow. In Appendix E.4, we show the full corner plot in Figure E.3 and marginals of the `garch11` model in Figure E.4.

## 5 Discussion of limitations and future work

In this work, we introduced EigenVI, a new approach for score-based variational inference based on orthogonal function expansions. The score-based objective for EigenVI is minimized by solving an eigenvalue problem, and thus this framework provides an alternative to gradient-based methods for BBVI. Importantly, many computations in EigenVI can be parallelized with respect to the batch of

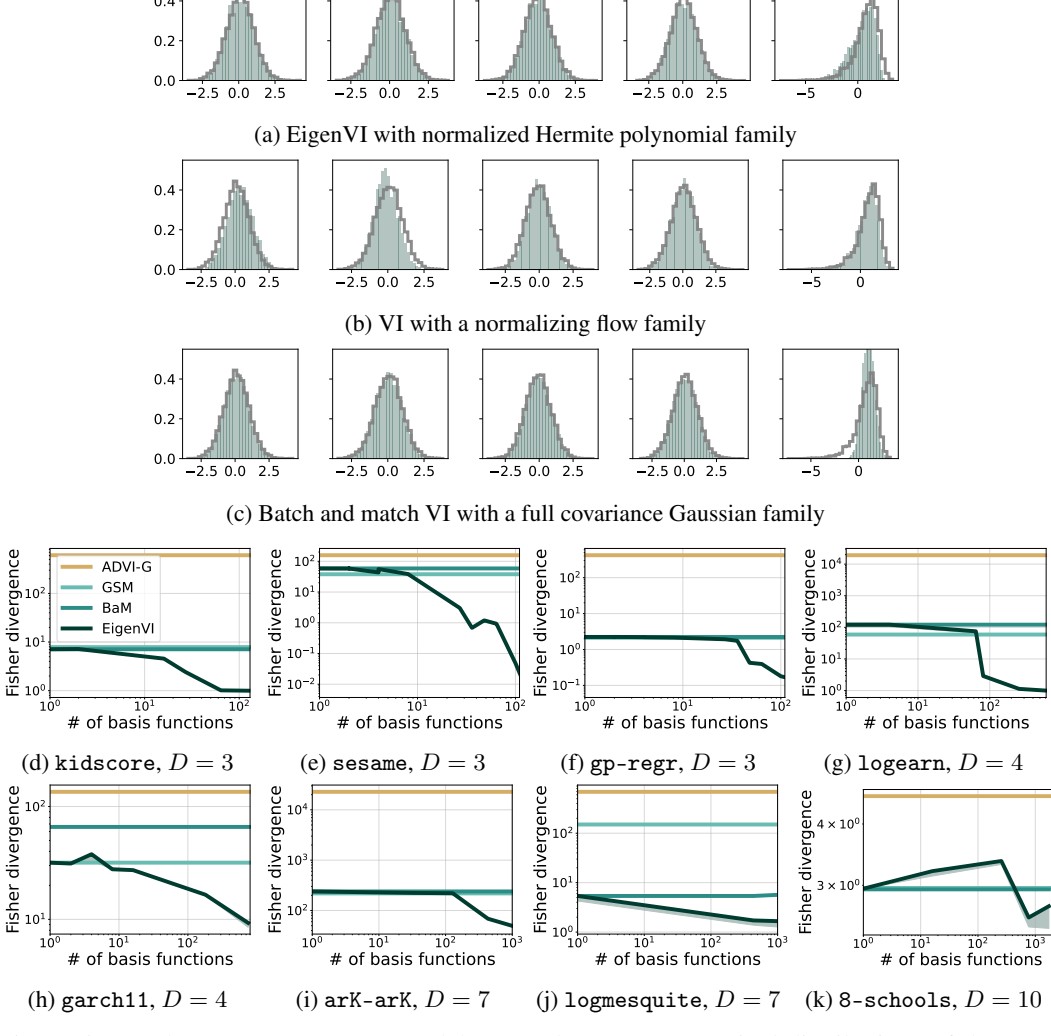

(a) EigenVI with normalized Hermite polynomial family

(b) VI with a normalizing flow family

(c) Batch and match VI with a full covariance Gaussian family

(d) `kidscore`, $D = 3$    (e) `sesame`, $D = 3$    (f) `gp-regr`, $D = 3$    (g) `logearn`, $D = 4$

(h) `garch11`, $D = 4$    (i) `arK-arK`, $D = 7$    (j) `logmesquite`, $D = 7$    (k) `8-schools`, $D = 10$

Figure 5: Results on `posteriordb` models. Top three rows: marginal distributions of the even dimensions from `8-schools`. Reference samples from HMC are outlined in gray, and the VI samples are in green. Bottom two rows: evaluation of methods with the (forward) Fisher divergence. The $x$-axis shows the number of basis functions, $K = \prod_d K_d$. Shaded regions represent standard errors computed with respect to 5 random seeds.

samples, unlike in iterative methods. We applied EigenVI to many synthetic and real-world targets, and these experiments show that EigenVI provides a principled way of improving upon Gaussian variational families.

Many future directions remain. First, the approach described in this paper relies on importance sampling, and thus it may benefit from more sophisticated methods for adaptive importance sampling. Second, it may be useful to construct variational families from different orthogonal function expansions. Our empirical study focused on the family built from normalized Hermite polynomials. But this family may require a very high-order expansion to model highly non-Gaussian targets, and such an expansion will be very expensive in high dimensions. Though this family was sufficient for many of the targets we simulated, others will be crucial for modeling highly non-Gaussian targets. Another direction is to develop variational families whose orthogonal function expansions scale more favorably with the dimension, perhaps by incorporating low rank structure in the target's covariance. Finally, it would be interesting to explore iterative versions of EigenVI in which each iteration solves a minimum eigenvalue problem on some subsample of data points. With such an approach, EigenVI could potentially be applied to very large-scale problems in Bayesian inference.

## Acknowledgments and Disclosure of Funding

We thank Bob Carpenter and Yuling Yao for helpful discussions and anonymous reviewers for their time and feedback on the paper. The Flatiron Institute is a division of the Simons Foundation. This work was supported in part by NSF IIS-2127869, NSF DMS-2311108, NSF/DoD PHY-2229929, ONR N00014-17-1-2131, ONR N00014-15-1-2209, the Simons Foundation, and Open Philanthropy.

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

## A  Sampling from orthogonal function expansions

In this appendix we show how to sample from a density on $\mathbb{R}^D$ constructed from a Cartesian product of orthogonal function expansions. Specifically, we assume that the density is of the form

$$q(z_1, z_2, \ldots, z_D) = \left( \sum_{k_1=1}^{K_1} \cdots \sum_{k_D=1}^{K_D} \alpha_{k_1 k_2 \ldots k_D} \phi_{k_1}(z_1) \phi_{k_2}(z_2) \cdots \phi_{k_D}(z_D) \right)^2, \tag{A.1}$$

where $\{\phi_k(\cdot)\}_{k=1}^{\infty}$ define a family of orthonormal functions on $\mathbb{R}$ and where the density is normalized by requiring that

$$\sum_{k_1 k_2 \ldots k_D} \alpha_{k_1 k_2 \ldots k_D}^2 = 1. \tag{A.2}$$

To draw samples from this density, we describe a sequential procedure based on inverse transform sampling. In particular, we obtain a sample $z \in \mathbb{R}^D$ by the sequence of draws

$$z_1 \sim q(z_1), \tag{A.3}$$
$$z_2 \sim q(z_2 | z_1), \tag{A.4}$$
$$\vdots$$
$$z_D \sim q(z_D | z_1, z_2, \ldots, z_{D-1}). \tag{A.5}$$

This basic strategy can also be used to sample from distributions whose domains are Cartesian products of different one-dimensional spaces.

In what follows, we first introduce a "core primitive" density, and we show how to sample efficiently from its distribution. We then show how the sampling procedure in Eqs. A.3–A.5 reduces to sampling from this core primitive; a key component of this procedure is the property of orthogonality, which helps facilitate the efficient computation of marginal distributions.

### Core primitive

First we describe the core primitive that we will use for each of the draws in Eqs. A.3–A.5. To begin, we observe the following: if $S$ is any positive semidefinite matrix with $\text{trace}(S) = 1$, then

$$\rho(\xi) = \sum_{k,\ell=1}^{K} S_{k\ell} \phi_k(\xi) \phi_\ell(\xi), \tag{A.6}$$

defines a normalized density over $\mathbb{R}$. In particular, since $S \succeq 0$, it follows that $\rho(\xi) \geq 0$ for all $\xi \in \mathbb{R}$, and since $\text{trace}(S) = 1$, it follows that

$$\int_{-\infty}^{\infty} \rho(\xi) \, d\xi = \sum_{k,\ell=1}^{K} S_{k\ell} \int_{-\infty}^{\infty} \phi_k(\xi) \phi_\ell(\xi) \, d\xi = \sum_{k,\ell=1}^{K} S_{k\ell} \delta_{kl} = \text{trace}(S) = 1. \tag{A.7}$$

The core primitive that we need is an efficient procedure to sample from a normalized density of this form. We will see later that all of the densities in Eqs. A.3–A.5 can be expressed in this form.

### Inverse transform sampling

Since the density in Eq. A.6 is one-dimensional, we can obtain the draw we need by inverse transform sampling. In particular, let $\mathcal{C}(\xi)$ denote the cumulative distribution function (CDF) associated with Eq. A.6, which is given by

$$\mathcal{C}(\xi) = \int_{-\infty}^{\xi} \rho(z) \, dz, \tag{A.8}$$

and let $\mathcal{C}^{-1}(\xi)$ denote the inverse CDF. Then at least in principle, we can draw a sample from $\rho$ by the two-step procedure

$$u \sim \text{Uniform}[0, 1], \tag{A.9}$$
$$\xi = \mathcal{C}^{-1}(u). \tag{A.10}$$

Next we consider how to implement this procedure efficiently in practice, and in particular, how to calculate the definite integral for the CDF in Eq. A.8. As shorthand, we define the doubly-indexed set of real-valued functions

$$\Phi_{k\ell}(\xi) = \int_{-\infty}^{\xi} \phi_k(z)\phi_\ell(z)\,dz. \tag{A.11}$$

It follows from orthogonality that $\Phi_{kl}(+\infty) = \delta_{kl}$ and from the Cauchy-Schwartz inequality that $|\Phi_{k\ell}(\xi)| \leq 1$ for all $\xi \in \mathbb{R}$. Our interest in these functions stems from the observation that

$$\mathcal{C}(\xi) = \sum_{k,\ell=1}^{K} S_{k\ell}\Phi_{kl}(\xi) = \text{trace}[S\Phi(\xi)], \tag{A.12}$$

so that if we have already computed the functions $\Phi_{k\ell}(\xi)$, then we can use Eq. A.12 to compute the CDF whose inverse we need in Eq. A.10. In practice, we can use numerical quadrature to pre-compute $\Phi_{k\ell}(\xi)$ for many values along the real line and then solve Eq. A.10 quickly by interpolation; that is, given $u$, we find $\xi$ satisfying $\text{trace}[S\Phi(\xi)] = u$. The result is an unbiased sample drawn from the density $\rho(\xi)$ in Eq. A.6.

**Sequential sampling**

Finally we show that each draw in Eqs. A.3–A.5 reduces to the problem described above. As in Section 2.1, we work out the steps specifically for an example in $D=3$, where we must draw the samples $z_1 \sim q(z_1)$, $z_2 \sim q(z_2|z_1)$ and $z_3 \sim q(z_3|z_1, z_2)$. This example illustrates all the ideas needed for the general case but with a minimum of indices.

Consider the joint distribution given by

$$q(z_1, z_2, z_3) = \left(\sum_{i=1}^{K_1}\sum_{j=1}^{K_2}\sum_{k=1}^{K_3} \beta_{ijk}\,\phi_i(z_1)\phi_j(z_2)\phi_k(z_3)\right)^2 \quad \text{where} \quad \sum_{ijk}\beta_{ijk}^2 = 1. \tag{A.13}$$

From this joint distribution, we can compute marginal distributions by integrating out subsets of variables, and each integration over $\mathbb{R}$ gives rise to a contraction of indices, as in Eq. 7, due to the property of orthogonality. In particular, expanding the square in Eq. A.13, we can write this joint distribution as

$$q(z_1, z_2, z_3) = \sum_{k,k'=1}^{K_3} \left[\sum_{i,i'=1}^{K_1}\sum_{j,j'=1}^{K_2} \beta_{ijk}\,\beta_{i'j'k'}\,\phi_i(z_1)\phi_{i'}(z_1)\phi_j(z_2)\phi_{j'}(z_2)\right] \phi_k(z_3)\phi_{k'}(z_3), \tag{A.14}$$

and we can then contract the index $k'$ when integrating over $z_3$, since $\int \phi_k(z_3)\phi_{k'}(z_3)dz_3 = \delta_{kk'}$.

In this way we find that the marginal distributions are

$$q(z_1, z_2) = \sum_{j,j'=1}^{K_2} \left[\sum_{i,i'=1}^{K_1}\sum_{k=1}^{K_3} \beta_{ijk}\beta_{i'j'k}\phi_i(z_1)\phi_{i'}(z_1)\right] \phi_j(z_2)\phi_{j'}(z_2), \tag{A.15}$$

$$q(z_1) = \sum_{i,i'=1}^{K_1} \left[\sum_{j=1}^{K_2}\sum_{k=1}^{K_3} \beta_{ijk}\beta_{i'jk}\right] \phi_i(z_1)\phi_{i'}(z_1). \tag{A.16}$$

Now note from the brackets in Eq. A.16 that this marginal distribution is already in the quadratic form of Eq. A.6 with coefficients

$$S_{ii'}^{(1)} = \sum_{j=1}^{K_2}\sum_{k=1}^{K_3} \beta_{ijk}\beta_{i'jk}. \tag{A.17}$$

From this first quadratic form, we can therefore use inverse transform sampling to obtain a draw $z_1 \sim q(z_1)$.

Next we consider how to sample from the conditional $q(z_2|z_1) = q(z_1, z_2)/q(z_1)$. Again, from the brackets in Eq. A.15, we see that this conditional distribution is also in the quadratic form of Eq. A.6 with coefficients

$$S_{jj'}^{(2)} = \frac{\sum_{i,i'=1}^{K_1} \sum_{k=1}^{K_3} \beta_{ijk} \beta_{i'j'k} \phi_i(z_1) \phi_{i'}(z_1)}{q(z_1)}. \tag{A.18}$$

From this second quadratic form, we can therefore use inverse transform sampling to obtain a draw $z_2 \sim q(z_2|z_1)$. Finally, we consider how to sample from $q(z_3|z_1, z_2) = q(z_1, z_2, z_3)/q(z_1, z_2)$. From Eq. A.14, we see that this conditional distribution is also in the quadratic form of Eq. A.6 with coefficients

$$S_{kk'}^{(3)} = \frac{\sum_{i,i'=1}^{K_1} \sum_{j,j'=1}^{K_2} \beta_{ijk} \beta_{i'j'k'} \phi_i(z_1) \phi_{i'}(z_1) \phi_j(z_2) \phi_{j'}(z_2)}{q(z_1, z_2)} \tag{A.19}$$

From this third quadratic form, we can therefore use inverse transform sampling to obtain a draw $z_3 \sim q(z_3|z_1, z_2)$. Finally, from the sums in Eq. A.19, we see that the overall cost of this procedure is $\mathcal{O}(K_1^2 K_2^2 K_3^2)$, or quadratic in the total number of basis functions.

# B  Calculation of moments

In this appendix we show how to calculate the low-order moments of a density constructed from the Cartesian product of orthogonal function expansions. In particular, we assume that the density is over $\mathbb{R}^D$ and of the form

$$q(z_1, z_2, \ldots, z_D) = \left( \sum_{k_1=1}^{K_1} \cdots \sum_{k_D=1}^{K_D} \alpha_{k_1 k_2 \ldots k_D} \phi_{k_1}(z_1) \phi_{k_2}(z_2) \cdots \phi_{k_D}(z_D) \right)^2, \tag{B.1}$$

where $\{\phi_k(\cdot)\}_{k=1}^{\infty}$ are orthogonal functions on $\mathbb{R}$ and where the coefficients are properly normalized so that the density integrates to one. For such a density, we show that the calculation of first and second-order moments boils down to evaluating *one-dimensional* integrals of the form

$$\mu_{ij} = \int_{-\infty}^{\infty} \phi_i(z) \phi_j(z) \, z \, dz, \tag{B.2}$$

$$\nu_{ij} = \int_{-\infty}^{\infty} \phi_i(z) \phi_j(z) \, z^2 \, dz. \tag{B.3}$$

We also show how to evaluate these integrals specifically for the orthogonal family of weighted Hermite polynomials.

First we consider how to calculate moments such as $\mathbb{E}_q[z_d^p]$, where $p \in \{1, 2\}$, and without loss of generality we focus on calculating $\mathbb{E}_q[z_1^p]$. We start from the joint distribution in Eq. B.1 and proceed by marginalizing over the variables $(z_2, z_3, \ldots, z_D)$. Exploiting orthogonality, we find that

$$\mathbb{E}_q[z_1^p] = \int q(z_1, z_2, \ldots, z_D) \, z_1^p \, dz_1 \, dz_2 \, \ldots dz_D, \tag{B.4}$$

$$= \int \left( \sum_{k_1=1}^{K_1} \cdots \sum_{k_D=1}^{K_D} \alpha_{k_1 k_2 \ldots k_D} \phi_{k_1}(z_1) \phi_{k_2}(z_2) \cdots \phi_{k_D}(z_D) \right)^2 z_1^p \, dz_1 \, dz_2 \, \ldots dz_D, \tag{B.5}$$

$$= \sum_{k_1, k_1'=1}^{K_1} \left[ \sum_{k_2=1}^{K_2} \cdots \sum_{k_D=1}^{K_D} \alpha_{k_1 k_2 \ldots k_D} \alpha_{k_1' k_2 \ldots k_D} \right] \int \phi_{k_1}(z_1) \phi_{k_1'}(z_1) \, z_1^p \, dz_1. \tag{B.6}$$

We can rewrite this expression more compactly as a quadratic form over integrals of the form in Eqs. B.2–B.3. To this end, we define the coefficients

$$A_{ij} = \sum_{k_2=1}^{K_2} \cdots \sum_{k_D=1}^{K_D} \alpha_{ik_2 \ldots k_D} \alpha_{jk_2 \ldots k_D}, \tag{B.7}$$

which simply encapsulate the bracketed term in Eq. B.6. Note that there are $K_1^2$ of these coefficients, each of which can be computed in $\mathcal{O}(K_2 K_3 \ldots K_D)$. With this shorthand, we can write

$$\mathbb{E}_q[z_1] = \sum_{i,j=1}^{K_1} A_{ij} \mu_{ij}, \tag{B.8}$$

$$\mathbb{E}_q[z_1^2] = \sum_{i,j=1}^{K_1} A_{ij} \nu_{ij}, \tag{B.9}$$

where $\mu_{ij}$ and $\nu_{ij}$ are the integrals defined in Eqs. B.2–B.3. Thus the problem has been reduced to a weighted sum of one-dimensional integrals.

A similar calculation gives the result we need for correlations. Again, without loss of generality, we focus on calculating $\mathbb{E}_q[z_1 z_2]$. Analogous to Eq. B.7, we define the tensor of coefficients

$$B_{ijk\ell} = \sum_{k_3=1}^{K_3} \cdots \sum_{k_D=1}^{K_D} \alpha_{ikk_3 \ldots k_D} \alpha_{j\ell k_3 \ldots k_D}, \tag{B.10}$$

which arises from marginalizing over the variables $(z_3, z_4, \ldots, z_D)$. There are $K_1^2 K_2^2$ of these coefficients, each of which can be computed in $\mathcal{O}(K_3 K_4 \ldots K_D)$. With this shorthand, we can write

$$\mathbb{E}_q[z_1 z_2] = \sum_{i,j=1}^{K_1} \sum_{k,\ell=1}^{K_2} B_{ijk\ell} \mu_{ij} \mu_{k\ell}. \tag{B.11}$$

where $\mu_{ij}$ is again the integral defined in Eq. B.2). Thus the problem has been reduced to a weighted sum of (the product of) one-dimensional integrals.

Finally, we show how to evaluate the integrals in Eqs. B.2–B.3 for the specific case of orthogonal function expansions with weighted Hermite polynomials; similar computations apply in the case of Legendre polynomials. Recall in this case that

$$\phi_{k+1}(z) = \left( \sqrt{2\pi} k! \right)^{-\frac{1}{2}} \left( e^{-\frac{1}{2} z^2} \right)^{\frac{1}{2}} \mathrm{H}_k(z), \tag{B.12}$$

where $\mathrm{H}_k(z)$ are the *probabilist's* Hermite polynomials given by

$$\mathrm{H}_k(z) = (-1)^k e^{\frac{z^2}{2}} \frac{d^k}{dz^k} \left[ e^{-\frac{z^2}{2}} \right]. \tag{B.13}$$

To evaluate the integrals for this particular family, we can exploit the following recursions that are satisfied by Hermite polynomials:

$$H_{k+1}(z) = z H_k(z) - H_k'(z), \tag{B.14}$$
$$H_k'(z) = k H_{k-1}(z). \tag{B.15}$$

Eliminating the derivatives $H_k'(z)$ in Eqs. B.14–B.15, we see that $z H_k(z) = H_{k+1}(z) + k H_{k-1}(z)$. We can then substitute Eq. B.12 to obtain a recursion for the orthogonal basis functions themselves:

$$z\phi_k(z) = \sqrt{k}\phi_{k+1}(z) + \sqrt{k-1}\phi_{k-1}(z). \tag{B.16}$$

With the above recursion, we can now read off these integrals from the property of orthogonality. For example, starting from Eq. B.2, we find that

$$\mu_{ij} = \int_{-\infty}^{\infty} \phi_i(z)\phi_j(z) z \, dz, \tag{B.17}$$

$$= \int_{-\infty}^{\infty} \phi_i(z) \left[ \sqrt{j}\phi_{j+1}(z) + \sqrt{j-1}\phi_{j-1}(z) \right] dz, \tag{B.18}$$

$$= \delta_{i,j+1}\sqrt{j} + \delta_{i,j-1}\sqrt{i}, \tag{B.19}$$

where $\delta_{ij}$ is the Kronecker delta function. Next we consider the integral in Eq. B.3, which involves a power of $z^2$ in the integrand. In this case we can make repeated use of the recursion:

$$\nu_{ij} = \int_{-\infty}^{\infty} \phi_i(z)\phi_j(z)\, z^2\, dz, \tag{B.20}$$

$$= \int_{-\infty}^{\infty} \left[ \sqrt{i}\phi_{i+1}(z) + \sqrt{i-1}\phi_{i-1}(z) \right] \left[ \sqrt{j}\phi_{j+1}(z) + \sqrt{j-1}\phi_{j-1}(z) \right] dz, \tag{B.21}$$

$$= \delta_{ij} \left[ \sqrt{ij} + \sqrt{(i-1)(j-1)} \right] + \delta_{i-1,j+1}\sqrt{j(j+1)} + \delta_{j-1,i+1}\sqrt{i(i+1)}. \tag{B.22}$$

Note that the matrices in Eq. B.19 and Eq. B.22 can be computed for whatever size is required by the orthogonal basis function expansion in Eq. B.1. Once these matrices are computed, it is a simple matter of substitution[1] to compute the moments $\mathbb{E}_q[z_1]$, $\mathbb{E}_q[z_1^2]$, and $\mathbb{E}_q[z_1 z_2]$ from Eqs. B.8–B.9 and Eq. B.11. Finally, we can compute other low-order moments (such as $\mathbb{E}_q[z_5]$ or $\mathbb{E}_q[z_3 z_7]$) by an appropriate permutation of indices.

## C  Eigenvalue problem

In this appendix we show in detail how the optimization for EigenVI reduces to a minimum eigenvalue problem. In particular we prove the following.

**Lemma C.1.** Let $\{\phi_k(z)\}_{k=1}^{\infty}$ be an orthogonal function expansion, and let $q \in \mathcal{Q}_K$ be the variational approximation parameterized by

$$q(z) = \left[ \sum_{k=1}^{K} \alpha_k \phi_k(z) \right]^2, \tag{C.1}$$

where the weights satisfy $\sum_{k=1}^{K} \alpha_k^2 = 1$, thus ensuring that the distribution is normalized. Suppose furthermore that $q$ is chosen to minimize the empirical estimate of the Fisher divergence given, as in eq. (10), by

$$\widehat{\mathcal{D}}_{\pi}(q, p) = \sum_{b=1}^{B} \frac{q(z^b)}{\pi(z^b)} \left\| \nabla \log q(z^b) - \nabla \log p(z^b) \right\|^2.$$

Then the optimal variational approximation $q$ in this family can be computed by solving the minimum eigenvalue problem

$$\min_{q \in \mathcal{Q}_K} \left[ \widehat{\mathcal{D}}_{\pi}(q, p) \right] = \min_{\|\alpha\|=1} \alpha^{\top} M \alpha =: \lambda_{\min}(M), \tag{C.2}$$

where $M$ is given in Eq. 13 and $\alpha = [\alpha_1, \ldots, \alpha_K] \in \mathbb{R}^K$. The optimal weights $\alpha$ are given (up to an arbitrary sign) by the corresponding eigenvector of this minimal eigenvalue.

*Proof.* The scores of $q$ in this variational family are given by

$$\nabla \log q(z^b) = \frac{2 \sum_k \alpha_k \nabla \phi_k(z^b)}{\sum_k \alpha_k \phi_k(z^b)}.$$

---

[1]With further bookkeeping, one can also exploit the *sparsity* of $\mu_{ij}$ and $\nu_{ij}$ to derive more efficient calculations of these moments.

Substituting the above into the empirical divergence, we find that

$$\widehat{\mathscr{D}}_\pi(q,p) = \sum_{b=1}^B \frac{q(z^b)}{\pi(z^b)} \left\| \nabla \log q(z^b) - \nabla \log p(z^b) \right\|^2$$

$$= \sum_{b=1}^B \frac{\left( \sum_k \alpha_k \phi_k(z^b) \right)^2}{\pi(z^b)} \left\| \frac{2 \sum_k \alpha_k \nabla \phi_k(z^b)}{\sum_k \alpha_k \phi_k(z^b)} - \nabla \log p(z^b) \right\|^2$$

$$= \sum_{b=1}^B \frac{1}{\pi(z^b)} \left\| 2 \sum_k \alpha_k \nabla \phi_k(z^b) - \left[ \sum_k \alpha_k \phi_k(z^b) \right] \nabla \log p(z^b) \right\|^2$$

$$= \sum_{b=1}^B \frac{1}{\pi(z^b)} \left\| \sum_k \alpha_k \left[ 2 \nabla \phi_k(z^b) - \phi_k(z^b) \nabla \log p(z^b) \right] \right\|^2$$

$$= \alpha^\top M \alpha,$$

where $M$ is given in (13) and $\alpha = [\alpha_1, \ldots, \alpha_K] \in \mathbb{R}^K$. Thus the optimal weights $\alpha$ are found by minimizing the quadratic form $\alpha^\top M \alpha$ subject to the constraint $\alpha^\top \alpha = 1$. Equivalently, a solution can be found by minimizing the Rayleigh quotient

$$\underset{v}{\mathrm{argmin}} \; \frac{v^\top M v}{v^\top v} \tag{C.3}$$

and setting $\alpha = v/\|v\|$. It then follows from the Rayleigh-Ritz theorem [8] for symmetric matrices that $\alpha$ is the eigenvector corresponding to the minimal eigenvalue of $M$, and this proves the lemma.

$\square$

# D  Practical considerations of EigenVI

## D.1  EigenVI vs gradient-based BBVI

Recall that EigenVI has two hyperparameters: the number of basis functions $K$ and the number of importance samples $B$. We note there is an important difference between these two hyperparameters and the learning rate in ADVI and other gradient-based methods. Here, as we use more basis functions and more samples, the resulting fit is a better approximation. So, we can increase the number of basis functions and importance samples until a budget is reached, or until the resulting variational approximation is a sufficient fit. On the other hand, tuning the learning rate in gradient-based optimization is much more sensitive because it cannot be too large or too small. If it is too large, ADVI may diverge. If the learning rate is too small, it may take too long to converge in which case it may exceed computational budgets.

Another fundamental difference in setting the number of basis functions as compared to the learning rate or batch size of gradient based optimization is that once we have evaluated the score of the target distribution for the samples, these same samples can be reused for solving the eigenvalue problem with any choice of the number of basis functions, as these tasks are independent. By contrast, in iterative BBVI, the optimization problem needs to be re-solved for every choice of hyperparameters, and the samples from different runs cannot be mixed together.

Furthermore, solving the eigenvalue problem is fast, and scores can be computed in parallel. In our implementation, we use off-the-shelf eigenvalue solvers, such as ARPACK [30] or Julia's eigenvalue decomposition function, `eigen`. In many problems with complicated targets, the main cost comes from gradient evaluation and not the eigenvalue solver.

## D.2  Choosing the number of samples $B$

Intuitively, if the target $p$ is in the variational family $\mathcal{Q}$ (i.e., it can be represented using an order-$K$ expansion), then we should choose the number of samples $B$ to roughly equal the number of basis function $K$. If $p$ is very different from $\mathcal{Q}$, we need more samples, and in our experiments, we use a multiple of the number of basis functions (say of order 10). As discussed before, once we have evaluated a set of scores, these can be reused to fit a larger number of basis functions.

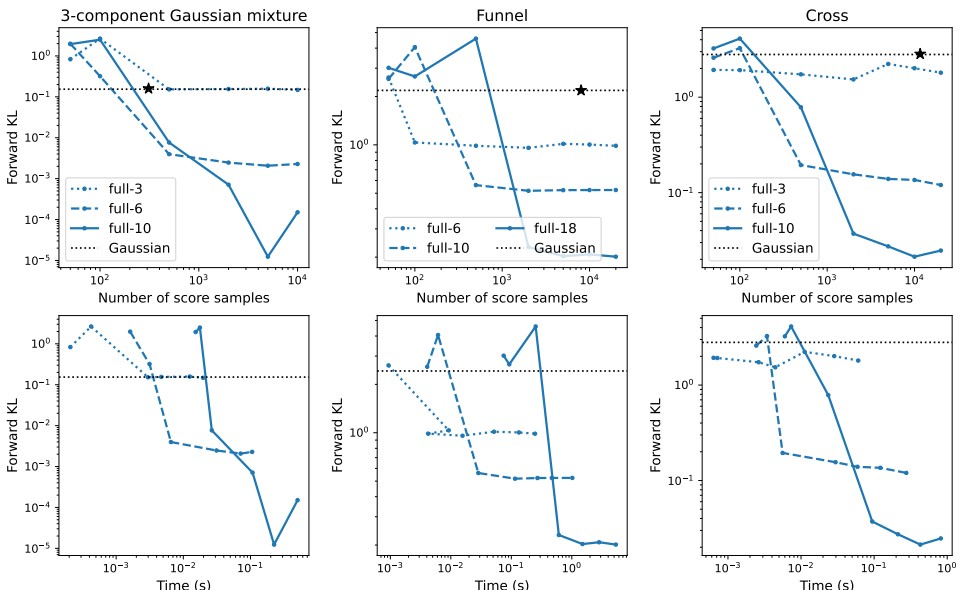

Figure E.1: We compare the number of score evaluations wallclock vs FKL divergence for the target distributions in Figure 3: the Gaussian mixture (column 1), the funnel (column 2), and the cross (column 3) distributions. The $K$ used for EigenVI is reported in each figure legend, where $K = K_1 K_2$. The black star denotes the number of gradient evaluations for the Gaussian method.

# E   Additional experiments and details

## E.1   Computational resources

The experiments were run on a Linux workstation with a 32-core Intel(R) Xeon(R) w5-3435X processor and with 503 GB of memory. Experiments were run on CPU. In the sinh-arcsinh and posteriordb experiments, computations to construct the matrix $M$ were parallelized over 28 threads.

## E.2   2D synthetic targets

We considered the following synthetic 2D targets:

- **3-component Gaussian mixture:**

$$p(z) = 0.4\mathcal{N}(z \,|\, [-1, 1]^\top, \Sigma) + 0.3\mathcal{N}(z \,|\, [1.1, 1.1]^\top, 0.5I) + 0.3\mathcal{N}(z \,|\, [-1, -1]^\top, 0.5I),$$

  where we define $\Sigma = \begin{bmatrix} 2 & 0.1 \\ 0.1 & 2 \end{bmatrix}$.

- **Funnel distribution with $\sigma^2 = 1.2$:**

$$p(z) = \mathcal{N}(z_1 \,|\, 0, \sigma^2)\mathcal{N}(z_2 \,|\, 0, \exp(z_1/2)).$$

- **Cross distribution:**

$$p(z) = \tfrac{1}{4}\mathcal{N}(z \,|\, [0, 2]^\top, \Sigma_1) + \tfrac{1}{4}\mathcal{N}(z \,|\, [-2, 0]^\top, \Sigma_2) + \tfrac{1}{4}\mathcal{N}(z \,|\, [2, 0]^\top, \Sigma_2) + \tfrac{1}{4}\mathcal{N}(z \,|\, [0, -2]^\top, \Sigma_1),$$

  where $\Sigma_1 = \begin{bmatrix} 0.15^{0.9} & 0 \\ 0 & 1 \end{bmatrix}$ and $\Sigma_2 = \begin{bmatrix} 1 & 0 \\ 0 & 0.15^{0.9} \end{bmatrix}$.

These experiments were conducted without standardization with a Gaussian VI estimate. The EigenVI proposal distribution $\pi$ used was a uniform$([-9, 9])$ distribution.

In Figure E.2, we run EigenVI for increasing numbers of importance samples $B$ and report the resulting forward KL divergence. The blue curves denote variational families with different $K_1 =$

$K_2 = K$ values used, i.e., 3, 6, and 10 (resulting in a total number of basis functions of $3^2$, $6^2$, and $10^2$). In the bottom row of the plot, we also show wall clock timings (computed without parallelization) to show how the cost grows with the increase in the number of basis functions and importance samples. The horizontal dotted line denotes the result from batch and match VI, which fits a Gaussian via score matching; here a batch size of 16 was used and a learning rate of $\lambda_t = \frac{BD}{t+1}$.

The black star denotes the number of score evaluations used by the Gaussian VI method.

### E.3 Sinh-arcsinh targets

The sinh-arcsinh normal distribution [19, 20] has parameters $s \in \mathbb{R}^D, \tau \in \mathbb{R}^D_+, \Sigma \in \mathbb{S}_{++}$; it is induced by transforming a Gaussian $Z_0 \sim \mathcal{N}(0, \Sigma)$ to $Z = \mathcal{S}_{s,\tau}(Z_0)$, where

$$\mathcal{S}_{s,\tau}(z) := [S_{s_1,\tau_1}(z_1), \ldots, S_{s_D,\tau_D}(z_D)]^\top, \quad S_{s_d,\tau_d}(z_d) := \sinh\left(\frac{1}{\tau_d}\sinh^{-1}(z_d) + \frac{s_d}{\tau_d}\right). \quad \text{(E.1)}$$

Here $s_d$ controls the amount of skew in the $d$th dimension, and $\tau_d$ controls the tail weight in that dimension. When $s_d=0$ and $\tau_d=1$ in all dimensions $d$, the distribution is Gaussian.

The sinh-arcsinh normal distribution has the following density:

$$p(z; s, \tau, \Sigma) = [(2\pi)^D|\Sigma|]^{-\frac{1}{2}}\prod_{d=1}^{D}\left\{(1+z_d^2)^{-\frac{1}{2}}\tau_d\, C_{s_d,\tau_d}(z_d)\right\}\exp\left(-\frac{1}{2}\mathcal{S}_{s,\tau}(z)^\top\Sigma^{-1}\mathcal{S}_{s,\tau}\right),$$
$$\text{(E.2)}$$

where we define the functions

$$C_{s_d,\tau_d}(z_d) := (1 + S_{s_d,\tau_d}^2(z))^{\frac{1}{2}}, \quad \text{(E.3)}$$

and

$$S_{s_d,\tau_d}(z_d) := \sinh(\tau_d\sinh^{-1}(z_d) - s_d), \quad S_{s,\tau}(z) = [S_{s_1,\tau_1}(z_1), \ldots, S_{s_D,\tau_D}(z_D)]^\top. \quad \text{(E.4)}$$

We constructed 3 targets in 2 dimensions and 3 targets in 5 dimensions, each with varying amounts of non-Gaussianity. The details of each target are below. In all experiments, EigenVI was applied with standardization, where a Gaussian was fit using batch and match VI with a batch size of 16 and a learning rate $\lambda_t = \frac{BD}{t+1}$.

For all experiments, we used a proposal distribution $\pi$ that was uniform on $[-5, 5]^2$.

**2D sinh-arcsinh normal experiment**  For $D = 2$ (Figure 4b), we consider the *slight skew and tails target* with parameters $s = [0.2, 0.2], \tau = [1.1, 1.1]$, the *more skew and tails target* with $s = [0.2, 0.5], \tau = [1.1, 1.1]$, and *the slight skew and heavier tails* with $s = [0.2, 0.2], \tau = [1.4, 1.1]$. Note that $s = [0, 0], \tau = [1, 1]$ recovers the multivariate Gaussian. These three target are visualized in Figure 4a.

**5D sinh-archsinh normal experiment**  We constructed three targets $P_1$ (slight skew and tails), $P_2$ (more skew and tails), and $P_3$ (slight skew and heavier tails) each with

$$\Sigma = \begin{bmatrix} 2.2 & 0.3 & 0 & 0 & 0.3 \\ 0.3 & 2.2 & 0 & 0 & 0 \\ 0 & 0 & 2.2 & 0.3 & 0 \\ 0 & 0 & 0.3 & 2.2 & 0 \\ 0.3 & 0 & 0 & 0 & 2.2 \end{bmatrix}. \quad \text{(E.5)}$$

The skew and tail weight parameters used were: $s_1 = [0., 0., 0.2, 0.2, 0.2]; \tau_1 = [1., 1., 1., 1., 1.1]$, $s_2 = [0.0, 0.0, 0.6, 0.4, -0.5]; \tau_2 = [1., 1., 1., 1., 1.1]$, and $s_3 = [0.2, 0.2, 0.2, 0.2, 0.2]; \tau_3 = [1.1, 1.1, 1., 1.4, 1.6]$. See Figure E.2 for a visualization of the marginals of each target distribution. In the second row, we show examples of resulting EigenVI fit (visualized using samples from $q$) from $B = 20,000$ and $K = 10$.

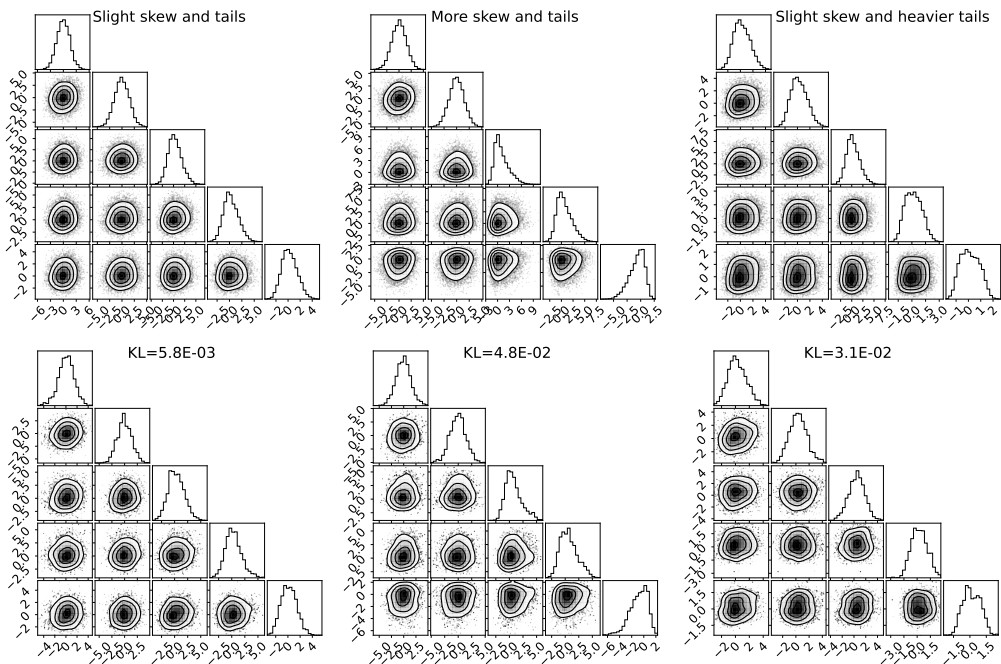

Figure E.2: Targets (top) for the 5D sinh-arcsinh normal distribution example and EigenVI fits (bottom) with the KL divergence in the figure title.

Table E.1: Summary of `posteriordb` models

| Name | Dimension | Model description |
| --- | --- | --- |
| `kidscore` | 3 | linear model with a Cauchy noise prior |
| `sesame` | 3 | linear model with uniform prior |
| `gp_regr` | 3 | Gaussian process regression with squared exponential kernel |
| `garch11` | 4 | generalized autoregressive conditional heteroscedastic model |
| `logearn` | 4 | log-log linear model with multiple predictors |
| `arK-arK` | 7 | autoregressive model for time series |
| `logmesquite` | 7 | multiple predictors log-log model |
| `8-schools` | 10 | non-centered hierarchical model for 8-schools |

### E.4 Posteriordb experiments

We consider 8 real data targets from `posteriordb`, a suite of benchmark Bayesian models for real data problems. In Table E.1, we summarize the models considered in the study. These target distributions are non-Gaussian, typically with some skew or different tails. To access the log target probability and their gradients, we used the BridgeStan library [42], which by default transforms the target to be supported on $\mathbb{R}^D$.

For all experiments, we fixed the number of importance samples to be $B = 40{,}000$; to construct the EigenVI matrix $M$, the computations were parallelized over the samples. These experiments were repeated over 5 random seeds, and we report the mean and standard errors in Figure 5; for lower dimensions, there was little variation between runs.

The target distributions were standardized using a Gaussian fit from score matching before applying EigenVI. In most cases, the proposal distribution $\pi$ was chosen to be uniform over $[-6, 6]^D$. For the models `8-schools`, which has a longer tail, we used a multivariate Gaussian proposal with zero mean and a scaled diagonal covariance $\sigma I$, with $\sigma = 3^2$.

For the Gaussian score matching (GSM) method [37], we chose a batch size of 16 for all experiments. We generally found the results were not too sensitive in comparison to other batch sizes of 4,8, and

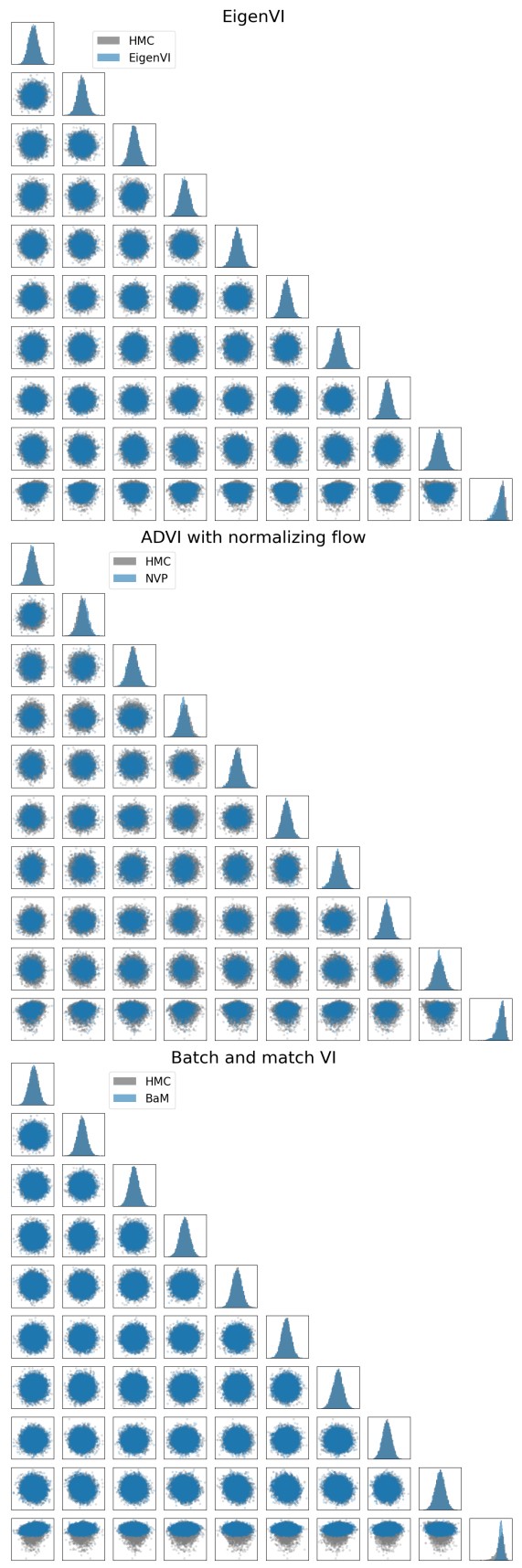

Figure E.3: Comparison of EigenVI, normalizing flow, and Gaussian score-based BBVI methods on 8schools.

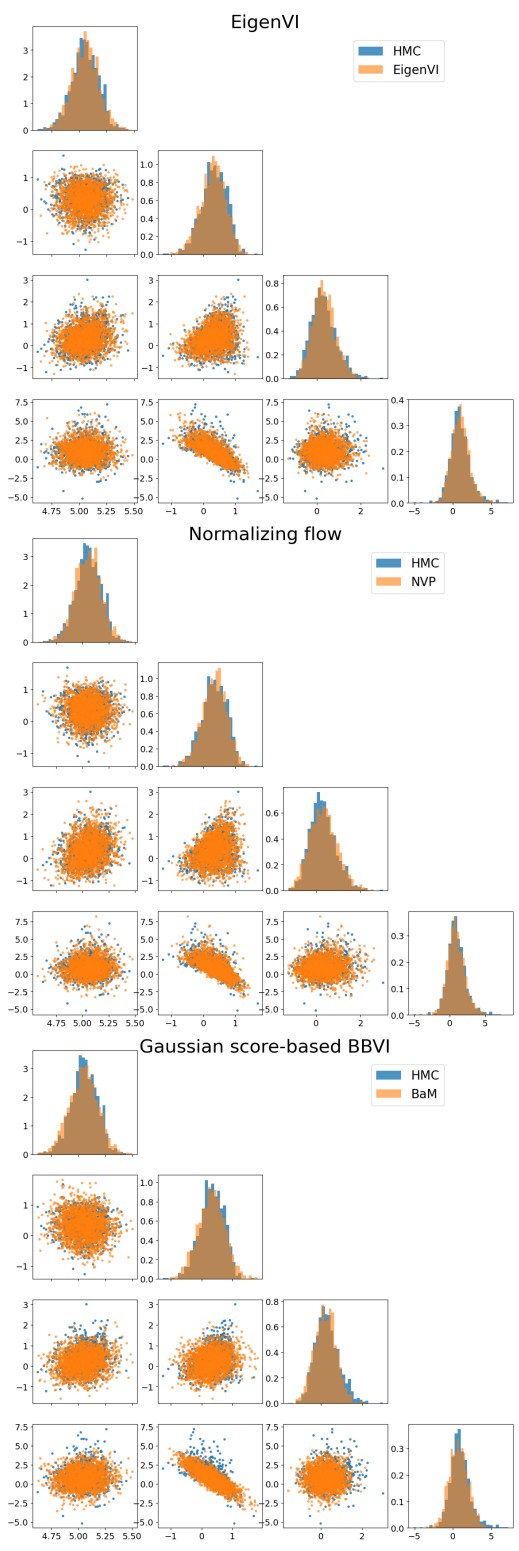

Figure E.4: Comparison of EigenVI, normalizing flow, and Gaussian score-based BBVI methods on `garch11`. Note that the Gaussian approximation over/underestimates the tails, while the more expressive families fit the tails better.

32. For the batch and match (BaM) method [6], we chose a batch size of 16. The learning rate was fixed at $\lambda_t = \frac{BD}{t+1}$, which was a recommended schedule for non-Gaussian targets.

For all ELBO optimization methods (full covariance Gaussian family and normalizing flow family), we used Adam to optimize the ELBO. We performed a grid search over the learning rate $0.01, 0.02, 0.05, 0.1$ and batch size $B = 4, 8, 16, 32$. For the normalizing flow model, we used a real NVP [12] with 8 layers and 32 neurons. We found empirically that the computational of the scores was unreliable [26, 49]; hence we do not show their Fisher divergence in Figure 5.

In Figure E.3 and Figure E.4, we show the corner plots that compare an EigenVI fit, a normalizing flow fit, and a Gaussian fit (BaM). In each plot, we plot the samples from the variational distribution against samples from Hamiltonian Monte Carlo. We observe that the two more expressive families EigenVI and the normalizing flow are able to model the tails of the distribution better than the Gaussian fit.

## F   Broader impacts

EigenVI adds to the literature on BBVI, which has been an important line of work for developing automated, approximate Bayesian inference methods. In terms of positive societal impacts, Bayesian models are used throughout the sciences and engineering, and advances in fast and automated inference will aid in advances in these fields. In terms of negative societal impacts, advances in BBVI could be used to train generative models with malicious or unintended uses.

