# OpenReview forum: "EigenVI: score-based variational inference with orthogonal function expansions"
_NeurIPS.cc/2024/Conference — NeurIPS 2024 spotlight_

### Official Review · Reviewer_JZVK · 2024-06-20

**Soundness:** 3
**Presentation:** 4
**Contribution:** 2
**Rating:** 5
**Confidence:** 4

**Summary:**

This work proposes a new variational family based on orthogonal function expansions from the quantum mechanics literature. Furthermore, the paper proposes to find the basis coefficients by solving a score matching problem, where the problem cleverly reduces to an eigenvalue problem. The overall approach, called EigenVI, is compared to recent score-matching-based Gaussian VI algorithms and BBVI.

**Strengths:**

* The idea of using orthogonal function expansions is new, although it reminds me of some other non-parametric/almost parametric variational families.
* The fact that the score-matching problems reduce to an eigenvalue problem is beautiful in its own right.
* The writing of the article is the best in my batch.

**Weaknesses:**

* While the authors are upright about the major limitation of the work, scaling with respect to dimensionality, this, unfortunately, strongly limits the practicality of the work. To me, for a VI algorithm to be useful, it should at least be scalable with respect to $d$, the dimensionality of the posterior, or $n$, the number of datapoints. Unfortunately, the proposed approach seems to achieve neither. In particular, the eigenvalue problem formulation is very cute but it isn't clear if subsampling results in an unbiased objective to be scalable with respect to $n$. (Please correct me if this is not the case!)
* The evaluation is not adequate for a work proposing a new variational family. The two baseline variational families considered here are the full-rank Gaussian variational family and normalizing flows. There is a rather long history of developing less-parametric variational families in VI. For instance, boosting VI [1], copulas [2,3], splines [4], semi-implicit families [5], mixtures [6], and many more. These works should be the main focus of the evaluation, not Gaussian VI.
* Unless I am not mistaken, the method suffers in the case the posterior is not *a priori* standardized. It doesn't seem like such standardization would be possible unless some preliminary inference has already been performed. This would defeat the point of using a VI algorithm. If one has already obtained preconditioners, why not just provide them to an MCMC algorithm instead?
* The paper doesn't discuss how to solve the eigenvalue problem. Given that we only need to solve a specific eigenvalue problem, there might be clever ways to solve it in a scalable way, maybe by using randomized linear algebra methods, which could technically be interesting.

Overall, I hope to see at least some evidence that the proposed methodology will eventually lead to a practical algorithm in the future. For instance, whether subsampling results in an unbiased/consistent algorithm. If this is the case, I would be happy to re-assess the paper.

## Additional Comments
* The Fisher divergence and the score-matching problem were introduced to the machine-learning community through the work of Hyvarinen [7], which isn't cited.
* Related works: A discussion on the literature of non-parametric/"less-parametric" VI algorithms seems necessary.
* Line 30 "One thread of BBVI research focuses on Gaussian variational approximations. (ADVI, BBVI)": I wouldn't quite agree with this statement since ADVI is applicable to any family where the reparameterization gradient is applicable. In fact, when normalizing flows are used with ELBO maximization, this is pretty much the same as ADVI. Furthermore, the original BBVI paper [8] requires conjugacy, meaning that one cannot simply use Gaussian variational families at all in a lot of cases.
* A lot of the references are citing the arXiv version. Berg et al. 2018 was published at UAI; Dinh et al. 2016 was published at ICLR 2017; Giordano et al. 2023 was published in JMLR; and etc. Please thoroughly review the reference section.
* Line 39, Section 2.1: Citing something when mentioning orthogonal functions would be better so that readers interested in the topic could have a look into more in-depth material.

## References
1. Campbell, T., & Li, X. (2019). Universal boosting variational inference. NeurIPS
2. Smith, M. S., & Loaiza-Maya, R. (2023). Implicit copula variational inference. Journal of Computational and Graphical Statistics, 32(3), 769-781.
3. Han, S., Liao, X., Dunson, D., & Carin, L. (2016, May). Variational Gaussian copula inference. In AISTATS.
4. Shao, Y., Shan, N. Y., & Feng, T. (2024, April). Nonparametric Automatic Differentiation Variational Inference with Spline Approximation. In AISTATS.
5. Yu, L., & Zhang, C. (2023). Semi-Implicit Variational Inference via Score Matching. arXiv preprint arXiv:2308.10014.
6. Lambert, M., Chewi, S., Bach, F., Bonnabel, S., & Rigollet, P. (2022). Variational inference via Wasserstein gradient flows. NeurIPS.
7. Hyvärinen, A., & Dayan, P. (2005). Estimation of non-normalized statistical models by score matching. Journal of Machine Learning Research, 6(4).
8. Ranganath, R., Gerrish, S., & Blei, D. (2014, April). Black box variational inference. In AISTATS.

**Questions:**

* In Line 103 to 105, it is stated that one can circumpass distributions with constrained supports. Given bijectors (ADVI), constrained supports are rarely a problem for VI algorithms these days. But dealing with *discrete support* is still a major challenge. Would there be a way to use the proposed method for problems with discrete support?
* Eq (10): Why is importance sampling used here when one can sample directly from $q$?

**Limitations:**

The paper would benefit from a dedicated limitation section.

---

> ### Author Rebuttal · Authors · 2024-08-07
>
> Thanks for your feedback on the paper. We discuss several of the points below.
>
> > 1. It isn't clear if subsampling results in an unbiased objective
>
> Any unbiased estimate of the gradient to our objective results in an unbiased estimate of the objective. And so EigenVI can scale to large n with subsampling.
>
> Let $p(z,x)$ be the target joint distribution between data $x$ and latent variables $z$.
>
> Let $g(z)$ be an unbiased estimate of the score $\nabla \log p(z,x)$, i.e., $\mathbb{E}[g(z) \mid z] = \nabla \log p(z,x)$.
>
> Expanding the Fisher divergence,
>
> $\mathbb{E}[\|\nabla \log q(z_b) - g(z_b)\|^2 \mid z_b] = \|\nabla \log q(z_b)\|^2 - 2 \langle \nabla \log q(z), \mathbb{E}[g(z_b) \mid z_b] \rangle + \mathbb{E}[\|g(z_b)\|^2 \mid z_b]$.
>
> Using the unbiased property, this becomes
>
> $\|\nabla \log q(z)\|^2 - 2 \langle \nabla \log q(z), \nabla \log p(z,x) \rangle + \mathbb{E}[\|g(z_b)\|^2 \mid z_b]$.
>
> So we get
>
> $\|\nabla \log q(z) - \nabla \log p(z,x)\|^2 + \mathbb{E}[\|g(z_b)\|^2 \mid z_b] - \|\nabla \log p(z,x)\|^2$.
>
> Thus, $\|\nabla \log q(z_b) - g(z_b)\|^2$ is an unbiased estimate of $\|\nabla \log q(z) - \nabla \log p(z,x)\|^2$ up to constants that do not depend on $q$. Because of this, we can use $g(z_b)$ as a drop-in replacement within the Fisher divergence within our method since
>
> $\min_{q \in \mathcal{Q}} \mathbb{E}\_{q(z)} [|\nabla \log q(z) - \nabla \log p(z,x)|^2] = \mathbb{E}_{q(z)}[|\nabla \log q(z) - g(z)|^2] + C.$
>
> To build this unbiased estimator of the score, we can use data subsampling as follows: suppose data is sampled i.i.d such that
>
> $p(z,x) = \prod_{i=1}^n p(z,x_i)$.
>
> Thus,
>
> $\nabla \log p(z,x) = \sum_{i=1}^n \nabla \log p(z,x_i)$.
>
> We can use this to get unbiased estimates of the full score by sampling the data $x_i$. For instance, by sampling $x_i \sim \frac{1}{n}$ we have that
>
> $\mathbb{E}[n \nabla \log p(z_b, x_i) \mid z_b] = \frac{1}{n} \sum_{j=1}^n n \nabla \log p(z_b, x_i) = \nabla \log p(z, x)$.
>
> We could also use any form of mini-batching over data to also build an unbiased estimate of the gradient.
>
> > 2. Scaling with dimension
>
> There are many directions for future work to help scale to higher dimensions. For instance, in many situations, one might expect non-Gaussianity on a low-dimensional subspace, and that the remaining dimensions can be suitably modeled with a Gaussian. This family is a special case of the orthonormal Hermite family considered in the paper and so is still an eigenvalue problem. Another direction is a low-rank approximation of the tensor – this family could be optimized with gradient-based methods. Finally, one may consider refining an existing basis to develop better basis functions.
>
> > 3. Why not use VI fit as a preconditioner for MCMC?
>
> As noted in above, MCMC struggles with targets with varying curvature (using a preconditioner does not help with multiscale distributions where curvature varies over different regions of the distribution, e.g., Neal's funnel). In the absence of pathological geometries, HMC can work very well in high-dimensions and so dimension itself doesn’t strike us as the relevant criterion to choose between VI or MCMC. Using VI with MCMC is an active research area, and this motivates developing new VI methods to benefit from their synergies.
>
> > 4. Comparing to non-parametric VI families
>
> We thank the referee for pointing to additional literature on non-parametric BBVI families. We will refer to these in the related works in the revised manuscript. However given the breadth of this literature, individual comparison with each of these is beyond the scope of this work.
>
> There are a few common elements to all these approaches, e.g., they all use gradient based optimization and hence are similarly sensitive as the baselines that we have considered. In addition, there are some individual challenges, e.g., mixtures are prone to mode collapse, and spline VI assumes a factorized posterior distribution.
>
> Given these considerations, we consider our baselines of full rank Gaussian VI and normalizing flows (NFs) to be adequate. Together, these are the most widely used variational families for BBVI, they encompass the two extremes on the spectrum of parametric vs non-parametric families, and especially NFs being universal approximators should typically encompass other variational families suggested by the reviewer.
>
> Finally, note that we do not suggest that EigenVI results in better final approximation than NFs, but instead our focus is introducing a new variational family and approach to inference which does not rely on typical iterative gradient based optimization and associated hyperparameter tuning (which is common to all aforementioned methods).
>
> > 5. The paper doesn't discuss how to solve the eigenvalue problem.
>
> Without further information on the scores of p, the resulting matrix in our eigenvalue problem is dense and unstructured, thus there is little we can leverage to design a specialized algorithm. In our experiments, we used off-the-shelf eigenvalue solvers. For typical problems, this is not a bottleneck (see the general response on the discussion for computational cost). We will add a discussion of this.
>
> > 6. References and related work
>
> We will expand our discussion of related nonparametric families, and we will add references to additional literature on orthogonal families and score matching.
>
> > 7. Given bijectors, constrained supports are rarely a problem. Would there be a way to use the proposed method for problems with discrete support?
>
> Transforming supports to be real-valued can sometimes make the problem more challenging (and adds an additional level of tuning to the problem). If we know that the variables are bounded, it may be more natural to model them in the original space. Regarding discrete support, this is an interesting direction for future work. With an appropriate basis set and replacing the scores with log ratios, one would still get an eigenvalue problem.

---

> > ### Comment · Reviewer_JZVK · 2024-08-07
> > **Response**
> >
> > Thank you for the detailed response.
> >
> > For the comment on unbiasedness, I should have phrased my original comment a little better. It is well known the score-matching objective itself is valid with subsampling. I was more concerned about the fact that it is unclear how to incorporate subsampling within the proposed algorithm in a consistent way.
> >
> > Nevertheless, I am convinced that the proposed methodology is technically interesting and will raise my score. However, I am still concerned that there isn't enough evidence that it would be practically applicable to larger-scale real-world problems. Thus, the paper feels somewhat less complete in its current state. For instance:
> >
> > > There are many directions for future work to help scale to higher dimensions. For instance, in many situations, one might expect non-Gaussianity on a low-dimensional subspace, and that the remaining dimensions can be suitably modeled with a Gaussian. This family is a special case of the orthonormal Hermite family considered in the paper and so is still an eigenvalue problem.
> >
> > Demonstrating this would have been much more convincing and made the work more complete.

---

> > > ### Author Response · Authors · 2024-08-12
> > >
> > > Thank you for the follow up &ndash; we now understand your question.
> > >
> > > Developing algorithms that support subsampling is an important line of future work, and we believe EigenVI develops some foundations that can be adapted to support subsampling. This may be achieved, for instance, with an iterative approach for updating variational parameters $\alpha_t$ that uses the score matching objective, where each iteration would involve solving for the minimum eigenvalue of a particular matrix&mdash;here each iteration could also take in scores from a subsampled batch of data points. However, a proper exploration of such a method warrants its own dedicated paper to explore a new algorithm (even before incorporating subsampling) and its properties, and we are planning to investigate these directions in future work. In the current work, we have maintained focus on introducing a novel variational family and non-iterative VI algorithm that we believe has its own merits.
> > >
> > > We think the question of subsampling will be of broad interest to other readers, and we will add a discussion to this end in the discussion & future work section of the revised paper. We thank the reviewer for bringing up this interesting point.

---

### Official Review · Reviewer_35nb · 2024-07-01

**Soundness:** 3
**Presentation:** 4
**Contribution:** 4
**Rating:** 7
**Confidence:** 4

**Summary:**

The authors propose EigenVI, a new method for black-box variational inference (BBVI).  The method uses the Fisher divergence (rather than the more typical reverse KL divergence) and a variational family based on orthogonal function expansions (which I've never seen used before).  Advantages of the approach include the ability to model complex distributions (multimodal, skewed, heavy-tailed, etc.) while also giving closed-form calculations for lower-order moments as well as easy posterior sampling.

**Strengths:**

The paper is exceptionally strong in originality, clarity, and significance.

For originality,  I have not previously seen a paper in machine learning using orthogonal function expansions to define a variational family.  The authors state that the approach comes from  quantum mechanics, an atypical source domain for a machine learning paper.  The construction is interesting, and normalizes elegantly.  Given the importance of orthogonal bases throughout mathematics, it is likely that this approach to variational inference will connect with many fields of academic inquiry.  Thus, this approach has the potential to greatly enrich research in variational inference.

For clarity, the exposition is exceptionally clear nearly everywhere (although Sec 2.3 could use a couple iterations of rewriting to streamline the narrative).   In particular, orthogonal function expansions and score-based divergences were introduced in an exceedingly clear (and interesting) manner.  The high-level advantages of their method vs. others in the literature were also exceedingly clearly presented.

For significance, the approach is broadly useful (at least for low-dimensional posteriors); this is inherited from the fact that BBVI is constructed so as to be applicable to a wide set of applications without requiring any pen-and-paper calculations.   Moreover, by avoiding gradient-based inference in favor of solving an eigenvalue problem, their method would additionally remove the common headache of having to specify or learn a learning rate.

**Weaknesses:**

Some weaknesses:

(1) As mentioned by the authors, the number of basis functions scales exponentially in the dimensionality of the posterior, and therefore so does inference time.   Given the restriction to low-dimensional posteriors, why use VI at all, rather than MCMC sampling?  It would have been nice to see this addressed in the exposition.  It would have also been nice to see an experimental comparison showing some advantage over using MCMC samplers.   For instance, do the authors expect a computational advantage of their approach over MCMC methods?  If so, it's hard to trust this since computation times are not given.

(2) Following up on the preceding point, it is completely unclear how long it takes to run this method, since no computation times are given relative to competitors.  On pp.9, the authors say, "In these examples, we standardize the target using either GSM or BaM before applying EigenVI; for this reason, we do not compare the costs of these methods to EigenVI."  I found this sentence quite puzzling.  Presumably the authors intend to argue that since EigenVI must take strictly longer than GSM or BaM,  there's not point in reporting the computation times.  If so, I don't see how the conclusion follows. A practitioner would want to know how MUCH additional compute is required to obtain any advantages.  Moreover, how does the overall (combined) computation time compare to other approaches, such as VI with a normalizing flow family or standard ADVI (Kucukelbir et al. 2017)?

(3) The method seems fairly complex to run, since a competitor BBVI method must be run simply as a preprocessing step to standardize the distribution.  (They use either Gaussian Score Matching or Batch-and-Match VI, per pp. 8.)

(4) A major selling point emphasized by the authors is that unlike gradient-based approaches, EigenVI does not require tunable hyper-parameters, such as learning rates or termination criteria.  However, EigenVI is clearly sensitive to (a) the number of basis functions and (b) the number of importance samples. Indeed, Figure 4 (c) demonstrates how inference can go catastrophically bad when the number of importance samples is not sufficiently large relative to the number of basis functions. This problem is reminiscent of the problem of a poorly chosen learning rate in standard ADVI (Kucukelbir et al., 2017).  Can the authors provide any guidance on how to choose a good number of basis functions for a given problem, and then, given that, how to choose a good number of importance samples?  A theoretical result would be nice, although admittedly it's unreasonable to expect that during the rebuttal period.  Do they at least have any practical insight to share?

(5) I feel that the authors undersell their own method by neglecting to reinforce the primary selling points in the experiments section.  For example, the authors mention that their approach allows sampling, closed-form moment calculations, and modeling of distributions with mixed support. However, none of these things  appear explicitly in the experimental section.   Similarly, the authors emphasize that the normalizing flow competitor is "often sensitive to the hyperparameters of the flow architecture, optimization algorithm, and parameters of the base distribution" (pp.6).  Can the authors demonstrate that the posterior approximation deteriorates along these dimensions more easily than their own approach deteriorates as a function of number of importance samples and basis functions?

**Questions:**

Clarifications:

(1) pp.6: What is the proposed advantage of EigenVI over Zhang et al.'s energy-based approach with a closed-form solution?

(2) pp.7, Figure 4:  If the normalized Hermite polynomials contain Gaussian distributions as a "base" (i.e. when the number of basis functions is K=1), then why does EigenVI still beat ADVI when K=1?  I expected them to perform identically in this setting.

(3) pp.8: The authors state that they "standardize the target using either GSM or BaM before applying EigenVI".   Any reason that they pick those methods relative to others? Would they expect any arbitrary BBVI method (e.g. ADVI of Kucukelbir et al. 2017) to work just as well for this purpose?

(4) pp.9:  Should "this family is limited to target functions that are close to Gaussian" read "... that are close to mixtures of Gaussians"?  For example, the cross-distribution (Figure 3, row 3) does not look close to Gaussian to me, although it does look close to a mixture of Gaussians.

Typos:

(1) Abstract: "novel class variational approximations" should read "novel class of variational approximations".

**Limitations:**

Yes, authors describe limitations well.

---

> ### Author Rebuttal · Authors · 2024-08-07
>
> Thanks for your feedback; in the revised manuscript, we will add or expand our discussion to address several of the points you bring up, as detailed below and in the main rebuttal comment.
>
> > 1. A practitioner would want to know how MUCH additional compute is required to obtain any advantages.
>
> While the actual overhead in terms of computational time depends on the efficiency of implementation, we provide qualitative reasons to justify why this is competitive to other approaches:
>
> Standardization can use BBVI approaches (like ADVI, BaM or GSM), and hence the cost of standardization is the same as the cost of doing VI inference using these. Following this, we need to evaluate the scores for importance samples which can be fully parallelized and hence, depending on the implementation, adds minimal overhead in terms of computational time. The final step is to solve the eigenvalue problem which is independent of the dimensions and is fast in the regimes we consider (see general comment and attached pdf for details).
>
> > 2. EigenVI is clearly sensitive to (a) the number of basis functions and (b) the number of importance samples. [...] This problem is reminiscent of the problem of a poorly chosen learning rate in standard ADVI
>
> We will modify the text to clarify that we do have two hyper-parameters: the number of basis functions and the number of importance samples. But we do find there is an important difference between our two hyperparameters and the learning rate in ADVI and other gradient-based methods. As we use more basis functions and more samples, the resulting fitted $q$ is a better approximation. So, we can increase the number of basis functions and importance samples until a budget is reached, or until the resulting q is a good enough fit. Tuning the learning rate is much more sensitive because it cannot be too large or too small. If it is too large, ADVI may diverge. If it is too small, it may take too long to converge.
>
> Another fundamental difference in setting the # of bases as compared to the learning rate or batch size of gradient based optimization is that once we have evaluated the score of the target distribution for the samples, these same samples can be reused for solving the eigenvalue problem with any choice of the number of basis functions, as these tasks are independent. By contrast, in iterative BBVI, the optimization problem needs to be solved from scratch for every choice of hyperparameters, and the samples from different runs cannot be mixed together. Furthermore, solving the eigenvalue problem is fast, and scores can be computed in parallel.
>
> Finally, for choosing the proposal distribution, in the case of the Hermite family, we have reasonable defaults because we are standardizing and hence can use e.g. an isotropic Gaussian with a larger variance to have stable importance weights.
>
> > 3. Can the authors provide any guidance on how to choose a good number of basis functions for a given problem, and then, given that, how to choose a good number of importance samples? [...] Do they at least have any practical insight to share?
>
> In terms of practical insights, if the target p is in the variational family Q, then we need the number of samples = # of basis functions. If it is very different from Q, we need more samples, and we use a multiple of the number of basis functions (say of order 10). As discussed before, once we have evaluated a set of scores, these can be reused to fit a larger number of basis functions.
>
> > 4. Similarly, the authors emphasize that the normalizing flow competitor is "often sensitive to the hyperparameters of the flow architecture, optimization algorithm, and parameters of the base distribution"
>
> In the rebuttal pdf Fig. 1, we focus on a particular model considered in the paper, and we show the sensitivity of posterior estimates to the learning rate and the batch size for the normalizing flow and standard ADVI.
>
> > 5. What is the proposed advantage of EigenVI over Zhang et al.'s energy-based approach?
>
> The energy-based model is not normalized, and thus it cannot be evaluated or sampled from (without using MCMC). Indeed, their paper details an approach that uses HMC with the energy-based VI approach.
>
> > 6. If the normalized Hermite polynomials contain Gaussian distributions as a "base" (i.e. when the number of basis functions is K=1), then why does EigenVI still beat ADVI when K=1?
>
> In this case, the Gaussian fit by BaM is used as the standardizing distribution. Thus, K=1 can beat ADVI if the standardizing distribution converges faster / to a better distribution.
>
> > 7. The authors state that they "standardize the target using either GSM or BaM before applying EigenVI". Any reason that they pick those methods relative to others? Would they expect any arbitrary BBVI method (e.g. ADVI of Kucukelbir et al. 2017) to work just as well for this purpose?
>
> We discuss the motivation behind the standardization in Section 2.3; here you can pick your favorite Gaussian approximation to do this, e.g., a Laplace approximation or ADVI or directly estimating the target mean and covariance (we chose the score-based VI methods in the experiments because they require less tuning and often converge faster). In general, we expect a better estimate of the target mean and covariance to help due to the base distribution of the Hermite family being a standard Gaussian (and thus, lower-order expansions will be more accurate).
>
> > 8. Should "this family is limited to target functions that are close to Gaussian" read "... that are close to mixtures of Gaussians"?
>
> We agree this sentence could be more clear, and we’ll modify it. What we meant was that to get to higher-dimensional spaces, we are limited to target functions that are close to Gaussian.
> For lower dimensional models (e.g., the 2D targets), we can afford to take higher-order expansions, and are able to model highly non-Gaussian distributions; we generally found this to be true for up to 3 or 4 dimensions.

---

> > ### Comment · Reviewer_35nb · 2024-08-13
> >
> > Thank you authors for your detailed response.  After reading the rebuttal and other reviews, I continue to find the paper both technically solid and innovative, and will maintain my positive score.

---

### Official Review · Reviewer_MKWn · 2024-07-08

**Soundness:** 3
**Presentation:** 3
**Contribution:** 3
**Rating:** 7
**Confidence:** 5

**Summary:**

This paper proposes EigenVI, which uses orthonormal distribution functions as the basis of the variational distribution family $q$. Then, minimizing the difference between $q$ and the target distribution $p$ via minimizing the Fisher divergence (2-norm score distance) is turned into an eigen-decomposition problem. Authors show that EigenVI is very competitive in distribution approximation compared with alternatives.

**Strengths:**

* The presentation is clear. The math is solid.
* The proposed EigenVI is very intuitive and reasonable.
* The low-order basis and higher-order bases have their own interpretation of the distribution's feature, which is desirable.
* Experiments are extensive, including synthetic and real-world data.

**Weaknesses:**

See Questions.

**Questions:**

* Eq. 9, should be $\mathrm{d} z$? Line 130, the score of the target is not equal to the gradient of the log joint, but up to a scaling?
* Could the authors explain a bit more why centering the distribution is very important for better approximation?
* Is there any guidance about how to choose a good proposal distribution? How do authors choose their proposal distributions used in the experiments? Is the choice of the proposal distribution very significant or quite robust?
* Eq. 16 seems a bit weird, especially the term $L_z(x_{1:N})$. Besides, it is better to clearly state whether $x_{1:N}$ are independent samples or a sequence, and how it relates to a single $z$.
* Why is forward KL used in the whole paper rather than the commonly used backward KL as in ELBO?
* How about the actual running time comparison on synthetic and real-world experiments? I understand that theoretically comparison is hard because eigenVI's complexity is w.r.t. number of basis (order) but the stochastic algorithm's complexity is mainly w.r.t. # epochs. However, solving large eigen-decomposition problems in practice might be very time-consuming. So it is very helpful to have an empirical understanding of the running time of the algorithm.

**Limitations:**

/

---

> ### Author Rebuttal · Authors · 2024-08-07
>
> Thanks for your feedback on the paper. In the revised manuscript, we will work on clarifying the following points.
>
> > 1. Should Equation 9 be dz?
>
> We could alternatively write $q(z) dz$ here (with the appropriate adjustments of $q(z)$ and $p(z)$).
>
> > 2. Line 130: the score of the target is not equal to the gradient of the log joint but up to a scaling?
>
> The log target is equal to the log joint + constant, and so the constant disappears when we take the gradient. Thus, $\nabla \log p(z) = \nabla \log \rho(z)$, where $\rho$ represents the unnormalized model (i.e., the joint in the context of Bayesian inference).
>
> > 3. On Equation 16 and the likelihood:
>
> This equation describes a general set of Bayesian problems from a set of benchmark models, and so we don’t a priori make assumptions on the conditional independence structure of the likelihood. We are happy to write $p(x_{1:N} | z)$ instead.
>
> > 4. Could the authors explain a bit more why centering the distribution is very important for better approximation?
>
> The standardization is used with the Hermite polynomial basis, which has a centered Gaussian as the base distribution. As we show in Figure 2, higher order expansions are needed to model distributions whose largest mode is shifted away from the origin. While this is fine in low dimensions, in order to handle higher dimensional targets, we apply the standardization technique as a way to reduce the order needed to model the target.
>
> > 5. Why is forward KL used rather than the commonly used backward KL as in ELBO?
>
> The ELBO is often used when comparing VI algorithms that use the ELBO / reverse KL as the objective for all VI algorithms. Here we are comparing VI algorithms with _different_ objectives, and so we chose an evaluation metric that was agnostic to the algorithms’ objectives.
>
> > 6. How to choose a good proposal distribution? How were they chosen in experiments?
>
> Please see our general response above for intuition behind choosing the proposal distribution. See Appendix E for the exact proposal distribution used for each experiment.
>
> > 7. How about the actual running time comparison on synthetic and real-world experiments? [...] Solving large eigen-decomposition problems in practice might be very time-consuming.
>
> We do not need to solve the full eigendecomposition problem, but only need the smallest eigenvalue pair for a KxK symmetric matrix, which can be a much easier problem to solve. As we mention in the general rebuttal comment, for 3000 basis functions, which is around the largest number of basis functions considered in experiments, this takes under a second. In a qualitative context, in the rebuttal pdf, we report the time it takes to run iterative BBVI methods, and we report the time needed for the eigenvalue solve. For instance, our implementation of ADVI takes over 12 seconds to converge in this example.
>
> Thus overall, we expect the overhead in terms of computational time due to solving the eigenvalue problem to be minimal.
>
> There are two other steps in (Hermite) EigenVI - standardization and evaluating scores of importance samples. While the actual computational time for these depends on the efficiency of implementation, expect this to be competitive to iterative BBVI approaches for two reasons -- 1) Standardization is done using BBVI approaches themselves (like ADVI, BaM or GSM), and hence the cost of standardization is the same as the cost of doing VI inference using these. 2) The scores for the importance samples can be evaluated in a fully parallelized manner and hence, depending on the implementation, adds minimal overhead in terms of computational time.
>
> Finally, to present a quantitative comparison, we previously included a running time comparison for the synthetic 2D experiments, see Appendix E (Figure 6 shows both # of gradient evaluations and wallclock time without parallelization). Here we did not need to use standardization, and so we indicate when EigenVI and the iterative method for the Gaussian fit use the same number of gradient evaluations.

---

> > ### Comment · Reviewer_MKWn · 2024-08-07
> >
> > Thanks for the responses. I will keep my positive score.

---

### Official Review · Reviewer_jsA6 · 2024-07-12

**Soundness:** 3
**Presentation:** 4
**Contribution:** 2
**Rating:** 6
**Confidence:** 4

**Summary:**

The paper proposes Eigen-VI, a black-box variational inference method that uses orthogonal function expansions to parameterize the variational distribution and uses Fisher divergence as an objective. The method does not require gradient-based optimization method and behaves well in a set of synthetic and real application experiments.

**Strengths:**

Strengths
* The paper is well-written and easy to follow.
* The proposed Eigen-VI is simple and easy to implement since the optimization problem can be solved by computing the minimum eigenvalue of a given matrix, and it also achieves comparable or better performances on hierarchical modeling benchmarks compared to other black-box variational inference methods.

**Weaknesses:**

Weaknesses
* Compared to other BBVI methods, Eigen-WI restricts the variational families to be $K^{\text {th}}$-order variational family(defined in eq. 2). When generalize to other variational families, gradient-free optimization in this paper may fail. Similarly, if variational distribution is set to be a linear combination of function basis, the optimization problem in other BBVI methods would also be much simpler. Therefore, the contribution of this paper may not be very significant.

* Different proposal distributions in importance sampling may affect the property of the matrix $M$, and further affect the numerical efficiency or stability of computing $\lambda_{\text{min}}(M)$. Adding some analysis regarding this matter would enhance the paper.

**Questions:**

* In the experiment part of this paper, why does the author only use the Hermite polynomials as the variational family? Is this because other variational family performs badly or have intractable issues?

**Limitations:**

See Weaknesses above.

---

> ### Author Rebuttal · Authors · 2024-08-07
>
> Thanks for your feedback on the paper. We discuss several of the comments and questions below.
>
> > 1. Compared to other BBVI methods, Eigen-VI restricts the variational families to be 𝐾th-order variational family(defined in eq. 2). When generalize to other variational families, gradient-free optimization in this paper may fail.
>
> As the reviewer points out, the EigenVI approach works for any orthogonal basis set. Importantly, this approach does not just work for a single variational family but a very large class of variational families. In particular, there are many orthogonal basis sets beyond the ones listed in the paper in Table 1. For instance, one may apply a procedure such as Gram-Schmidt to obtain such a set. Finally, new variational families can be formed by even mixing different basis sets. Thus, this approach opens the door to applying VI to many new types of families.
>
> > 2. Similarly, if variational distribution is set to be a linear combination of function basis, the optimization problem in other BBVI methods would also be much simpler.
>
> Classical BBVI methods also have assumptions that restrict the variational families. For instance, it is common that the family supports reparameterization (which is not true here).
>
> > 3. Different proposal distributions in importance sampling may affect the property of the matrix M, and further affect the numerical efficiency or stability of computing λmin(M). Adding some analysis regarding this matter would enhance the paper.
>
> In principle we agree that a poor proposal distribution might cause such numerical issues; however, in the current experiments, we do not observe this. Standardizing the distribution as we currently do it also helps here, as it provides a good proposal distribution which can easily be adjusted to have heavier tails if necessary. Furthermore, with enough samples, and since solving for minimum eigenvalues is typically so fast even for large matrices, we expect these issues to be less important. Experimenting with different proposal distributions is beyond the scope of this rebuttal, but we acknowledge the good point raised and will add a remark to this end in the revised manuscript on how, e.g., this could affect the spectral gap.
>
> > 4. In the experiment part of this paper, why does the author only use the Hermite polynomials as the variational family? Is this because other variational family performs badly or have intractable issues?
>
> The parameters for most experiments considered lie on an unconstrained scale where Hermite polynomials form a natural basis; the Hermite family can also be seen as a natural extension of non-Gaussianity.
>
> The EigenVI method itself applies to _any orthogonal basis set_; in Figure 1, we additionally show 1D target distributions and fitted distributions arising from Legendre polynomial and Fourier series expansions.

---

> > ### Comment · Reviewer_jsA6 · 2024-08-12
> >
> > I thank the author for the detailed response. Despite this, I think the weaknesses of this paper in my review still remain. Therefore, I would keep this score.

---

### Official Review · Reviewer_XdZT · 2024-07-13

**Soundness:** 4
**Presentation:** 3
**Contribution:** 3
**Rating:** 7
**Confidence:** 3

**Summary:**

The authors introduce a novel family of variational distributions based on orthogonal function expansions which is optimized by minimizing a Fisher divergence. They show that an unbiased importance sampling estimate of the divergence results in a quadratic form, which can be minimized by finding its smallest eigenvector.

**Strengths:**

The work is original and of high overall quality. The proposed variational family and training methodology is novel and of interest to the variational inference community. The empirical evaluation is fairly extensive and shows that the method works well for low dimensional problems. At the same time the authors are very forward with discussing the current limitations of their method which opens up interesting questions for future research.

**Weaknesses:**

As discussed by the authors, the work seems to be mostly limited by the fact that higher-order function expansions, which are needed to model non-Gaussianity, become exceedingly expensive in high dimensions as a result of the blow up of the size of the eigenvalue problem.

**Questions:**

- The authors describe a procedure to sample from the variational approximation, so I am not sure why importance sampling needed to estimate the Fisher divergence? Is it purely for computational reasons or am I missing something?
- How is the proposal distribution chosen that is used for importance sampling?

**Limitations:**

The authors have adequately addressed the limitations of their work.

---

> ### Author Rebuttal · Authors · 2024-08-07
>
> Thanks for your interest in the paper. We will revise the final version of the paper to make the following points more clear.
>
> > 1. The authors describe a procedure to sample from the variational approximation, so I am not sure why importance sampling is needed to estimate the Fisher divergence? Is it purely for computational reasons or am I missing something?
>
> First, because we are optimizing the empirical Fisher divergence with respect to $q$, we need the samples to be independent of $q$ (we cannot simultaneously sample from and optimize over $q$). In addition, as we show in Appendix D, the importance weights are important and lead to the form of the eigenvalue problem.
>
> The procedure to sample from $q$ is primarily used after the variational distribution has been fit to, for instance, approximate functionals of $p$.
>
> We will revise the paragraph starting from Line 133 to explain these points more thoroughly.
>
> > 2. How is the proposal distribution chosen that is used for importance sampling?
>
> Please see our general response above for intuition behind choosing the proposal distribution. See Appendix E for the exact proposal distribution used for each experiment.
>
> We will add a paragraph in the Section 2.3 that details this intuition behind choosing the proposal distribution.

---

> > ### Comment · Reviewer_XdZT · 2024-08-12
> > **Re: Rebuttal by Authors**
> >
> > Thanks for addressing my questions!
> >
> > After reading all reviews and corresponding rebuttals, I will maintain my positive score. I believe this is a solid paper!

---

### Author Rebuttal · Authors · 2024-08-07

We thank the reviewers for their time and engagement – we are pleased that they found the work to be highly original, well-written, and of interest to the variational inference community.

In this work, we propose a novel variational family and an efficient algorithm to fit it via score matching. Here variational distributions are obtained by _squaring_ a linear combination of orthogonal basis functions – we demonstrate the flexibility of this family using several basis sets (e.g., Hermite, Fourier series, and Legendre polynomials) in Figure 1, but note that our approach applies to _any orthogonal basis set_, paving the way for many other possible families. We show that minimizing an estimate of the Fisher divergence (which matches the scores of q and p) is equivalent to solving for the minimum eigenvalue of some matrix. EigenVI avoids gradient-based optimization, and the scores and the matrix of interest can be evaluated and formed in parallel.

_**After thorough consideration of reviewers’ comments, we will expand our discussion of several points in the manuscript. Our response includes additional plots of hyperparameters and times for baseline methods.**_

## Score-based VI vs sampling & MCMC [Reviewers 35nb, JZVK]

First, a score-based approach has potential advantages over a purely sampling-based approach that does not exploit the availability of scores. These advantages are perhaps most easily understood by considering the simple problem of estimating the mean and variance of a one-dimensional Gaussian distribution, $p(z)$. By drawing $n$ samples from $p$, one can estimate the mean and variance to accuracy $O(1/\sqrt{n})$. However, one can determine the mean and variance _exactly_ from the scores $d\log(p)/dz$ at just two samples $z_1$ and $z_2$, provided that $z_1\neq z_2$. Most settings are not this contrived, but the broader point remains: the scores of a target distribution $p$ provide a great deal of information to constrain the search for the best variational approximation $q$ in some parameterized family of tractable distributions. Methods that do not exploit this information will generally be at a considerable disadvantage to those that do.

In addition, reviewers have asked specifically about MCMC vs EigenVI. We agree MCMC may be applied to several of the examples in the paper and a comprehensive comparison with MCMC strikes us as an important direction for future work. However, most off-the-shelf algorithms, such as Stan’s adaptive Hamiltonian Monte Carlo, do not work well on targets with a varying curvature (e.g. funnel) or well separated modes (e.g. cross), even with preconditioners, in low or high dimensions. In all cases, tuning MCMC—and in particular (i) determining the length of the warmup and sampling phase of the Markov chain, and (ii) deciding on a number of chains, are non-trivial tasks which severely impact the reported performance. Therefore, benchmarking against MCMC requires careful study.

## Cost of EigenVI & solving the eigenvalue problem [Reviewers 35nb, MKWn, JZVK]

EigenVI needs to compute only the smallest eigenpair of a K x K symmetric matrix, where K is the # of basis functions, and not the full eigendecomposition. We use up to several thousand basis functions. In this regime, standard solvers for the minimum eigenvalue take under a second; e.g., ARPACK, a popular backend wrapped in Python, Julia, and Matlab that implements the restarted Lanczos method, takes 510 ms to find the smallest eigenvalue of a 3000 x 3000 randomly generated positive definite matrix. When the dimension K is very large, too large to store the matrix M, we can use iterative solvers for computing the smallest eigenvalue, e.g., gradient descent applied to the Rayleigh coefficient $\alpha^\top M \alpha/\alpha^\top\alpha $. Each iteration of gradient descent would require a matrix-vector product, and costs O(K^2). Furthermore, we do not need to explicitly form the matrix M, but need only implement the operation $\alpha \mapsto M \alpha,$ and thus never need to store the K x K matrix.

Finally, we note in many problems with complicated targets, the main cost comes from gradient evaluation and not the eigenvalue solver.

## Why use importance sampling if we can sample directly from q? [Reviewers XdZT, JZVK]

Because we are optimizing the empirical Fisher divergence with respect to q, we need the samples to be independent of q (we cannot simultaneously sample from and optimize over q, and it’s unclear how to apply reparameterization). In addition, the importance-sampled estimate leads to the quadratic form and thus the eigenvalue problem.

## Proposal dist. for EigenVI [Reviewers XdZT, jsA6, 35nb]

In the particular case of the Hermite variational family, we can first standardize the target distribution using the mean and covariance of a Gaussian fit. Thus, intuitively, we want a proposal that has heavier tails than a standard Gaussian. We found reasonable defaults to be centered distributions such as a uniform (if most of the target’s mass is between certain bounds) or multivariate Gaussian with long enough tails.

## EigenVI does not require standardization, but it helps [Reviewers 35nb, JZVK]

If the variables are bounded, a uniform distribution is a reasonable base distribution, and no standardization is needed.

For the Hermite family, we do not need standardization in the lower dimensions (no standardization is used for the 2D targets in the experiments). Standardization is used specifically with the Hermite family to help scale the method to larger dimensions, as the base distribution of the Hermite family is a standard Gaussian. (As we show in Fig. 2 for a 1D example, uncentered distributions may require more basis functions, so standardization helps to reduce the # of basis functions needed.) Given standardization with a Gaussian distribution, one can also view EigenVI as a post-processing step of a Gaussian BBVI (with any BBVI technique of choice--ADVI, BaM or GSM).

---

### Comment · Area_Chair_6Xii · 2024-08-07
**Rebuttal?**

Dear authors (and reviewers),

There were some positive reviews, but also one negative and one borderline review.  At the end of the day, we don't simply average scores, so this paper is not guaranteed acceptance, though it certainly has a good chance.

Hence I'd advise the authors to write rebuttals, at least to the two most negative/neutral reviews.

Reviewers, please take a look at each others' reviews and see if that changes your mind. As a reminder, the OpenReview platform allows us to have a dialogue, so you can post small questions to the authors and authors can respond.

Best regards

Area Chair

---

### Decision · Program_Chairs · 2024-09-25

**Decision:**

Accept (spotlight)

**Comment:**

This paper had 5 reviewers instead of the normal 4 (or minimum of 3), and yet all reviewers were unanimous: this is an original paper of high quality. The idea is novel, the evaluation is solid, and the paper is honest about shortcomings. The topic, Variational Inference, is of wide-ranging interest in ML.

Therefore I'm pleased to recommend acceptance.

For the camera-ready version, reviewer JZVK strongly wants the authors to do a thorough review of non-parametric variational families. Please make any other changes requested by the reviewers that have not already been addressed in the revisions.